# Reticulon-1 synthesis controls outgrowth and microtubule dynamics in injured cortical axons

Alejandro Luarte[1,2,3] , Javiera Gallardo[8] , Daniela Corvalán[1,2], Ankush Chakraborty[1,2] , Cláudio Gouveia Roque[5] , Francisca Bertin[11] , Carlos Contreras[1,2], Juan Pablo Ramírez[1,2], André Weber[20], Waldo Acevedo[7] , Werner Zuschratter[19,20], Rodrigo Herrera-Molina[9,10] , Úrsula Wyneken[1,2,3] , Andrea Paula-Lima[16,17], Tatiana Adasme-Rocha[18] , Jorge Toledo[21] , Rodrigo Vergara[22], Antonia Figueroa[1,2], Carolina González[15], Christian González-Billault[4,12,13,14], Ulrich Hengst[5,6] , Andrés Couve[4]

The regenerative potential of developing cortical axons depends on intrinsic mechanisms, such as axon-autonomous protein synthesis, that are still not fully understood. An emerging factor in this regenerative response is the bidirectional interplay between microtubule dynamics and the axonal ER. We hypothesize that locally synthesized ER proteins regulate microtubule dynamics and the regeneration of cortical axons. RNA data mining identified the ER-shaping protein Reticulon-1 as a relevant candidate across eight axonal transcriptomes. Using microfluidics, we show that axonal treatment with a small RNA against Reticulon-1 mRNA (Reticulon-1 knockdown) increases outgrowth of injured cortical axons while reducing their tubulin levels. We show by live-cell imaging that axonal Reticulon-1 knockdown increases microtubule growth rate in noninjured axons and restores this parameter after injury. Axonal inhibition of the microtubule-severing protein Spastin prevents the effects of Reticulon-1 knockdown over tubulin levels and outgrowth. We provide evidence that the Reticulon-1C isoform is synthesized within axons and attenuates Spastin-mediated microtubule severing. These findings support a model in which axonal protein synthesis regulates microtubule dynamics and axon outgrowth after injury.

## Introduction

Developing cortical axons can regenerate after injury but lose this capacity at later stages (Cooke et al, 2022). The transition from a regenerative to a nonregenerative state highlights the importance of identifying the factors that may underlie this shift. The regenerative potential of developing axons depends on extrinsic and intrinsic (cell-autonomous) factors that regulate their outgrowth (He & Jin, 2016). The structure of the axonal ER is a determinant intrinsic factor for regulating outgrowth during development and after injury (Deitch & Banker, 1993; Krijnse-Locker et al, 1995; Aridor & Fish, 2009; Nozumi et al, 2009; Rao et al, 2016; Petrova et al, 2020). Indeed, ER proteins are locally synthesized in axons and may contribute to regeneration (Willis et al, 2005; Pacheco et al, 2020). However, the role of axonal protein synthesis in shaping ER function after injury remains largely unexplored.

Unlike the ribosome-rich somato-dendritic ER, the structure of the axonal ER predominantly comprises continuous tubules of smooth ER, maintained by specific ER-shaping proteins (Voeltz et al, 2006; Wu et al, 2017). As such, the mostly tubular axonal ER controls calcium handling, lipid provision, and potentially transmembrane protein supply, which contribute to axon elongation (González et al, 2016; Cornejo et al, 2017; de Juan-Sanz et al, 2017; Luarte et al, 2018). More recently, a bidirectional relationship of the axonal ER with microtubule dynamics has emerged as critical for

---

[1]Faculty of Medicine, Universidad de los Andes, Santiago, Chile    [2]Program in Neuroscience, Center for Biomedical Research, and Innovation (CiiB), Universidad de los Andes, Santiago, Chile   [3]IMPACT, Center of Interventional Medicine for Precision and Advanced Cellular Therapy, Santiago, Chile   [4]Department of Neuroscience, Faculty of Medicine, Universidad de Chile, Santiago, Chile   [5]The Taub Institute for Research on Alzheimer's Disease and the Aging Brain, Columbia University Vagelos College of Physicians and Surgeons, New York, NY, USA   [6]Department of Pathology and Cell Biology, Columbia University Vagelos College of Physicians and Surgeons, New York, NY, USA   [7]Instituto de Química, Facultad de Ciencias, Pontificia Universidad Católica de Valparaíso, Valparaíso, Chile   [8]Centro de Medicina Regenerativa, Facultad de Medicina, Clínica Alemana-Universidad del Desarrollo, Santiago, Chile   [9]Centro Integrativo de Biología y Química Aplicada, Universidad Bernardo O'Higgins, Santiago, Chile   [10]Department of Pharmacology and Physiology, George Washington University, Washington, DC, USA   [11]Facultad de Medicina Veterinaria y Agronomía, Universidad de Las Américas, Santiago, Chile   [12]Department of Biology, Faculty of Sciences, Universidad de Chile, Santiago, Chile   [13]Public Nutrition Unit, Institute for Nutrition and Food Technologies, Universidad de Chile, Santiago, Chile   [14]The Buck Institute for Research on Aging, Novato, CA, USA   [15]Latin American Brain Health Institute (BrainLat), Universidad Adolfo Ibáñez, Santiago, Chile   [16]Biomedical Neuroscience Institute and Department of Neurosciences, Faculty of Medicine, Universidad de Chile, Santiago, Chile   [17]Institute for Research in Dental Sciences (ICOD), Faculty of Dentistry, Universidad de Chile, Santiago, Chile   [18]Oficina de Apoyo a la Investigación Clínica, Hospital Clínico Universidad de Chile, Santiago, Chile   [19]Leibniz Institute for Neurobiology, Magdeburg, Germany   [20]Photonscore GmbH, Magdeburg, Germany   [21]Health Sciences Department, Universidad de Aysén, Coyhaique, Chile   [22]Departamento de Kinesiología, Facultad de Artes y Educación Física, Universidad Metropolitana de Ciencias de la Educación, Ñuñoa, Chile

Correspondence: aluarte@uandes.cl

**Table 1. Analyzed publications examining the transcriptome of an axon-enriched neuronal fraction.**

| Study | Neuron cell type | Model to obtain axonal fraction |
|---|---|---|
| Saal et al (2014) | Mouse motor neurons | Spinal cord motor neurons cultured in microfluidic chambers (7 DIV). |
| Rotem et al (2017) | Mouse motor neurons | Spinal cord motor neurons cultured in porous membrane inserts (10 DIV). |
| Maciel et al (2018) | iPSC-derived human motor neurons | Motor neurons were cultured on porous membrane inserts (10 DIV) |
| Bigler et al (2017) | Glutamatergic neurons differentiated from human embryonic stem cells (hESCs) | hESCs differentiated for 59 DIV in microfluidic chambers. |
| Nijssen et al (2018) | Human and mouse motor neurons. | Spinal cord and human stem cell–derived motor neurons were cultured in microfluidic chambers (7 and 14 DIV, respectively). |
| Farias et al (2020) | Rat mature myelinated motor neurons | Axoplasm microdissection of motor axons (ex vivo). |
| Gumy et al (2011) | Rat embryonic/adult sensory neurons from dorsal root ganglion (DRG) | Primary neurons were cultured in microfluidic chambers (4 DIV). |
| Taylor et al (2009) | Uninjured and regenerating rat cortical neurons | Primary neurons cultured in microfluidic chambers (10 DIV). |

DIV, days in vitro.

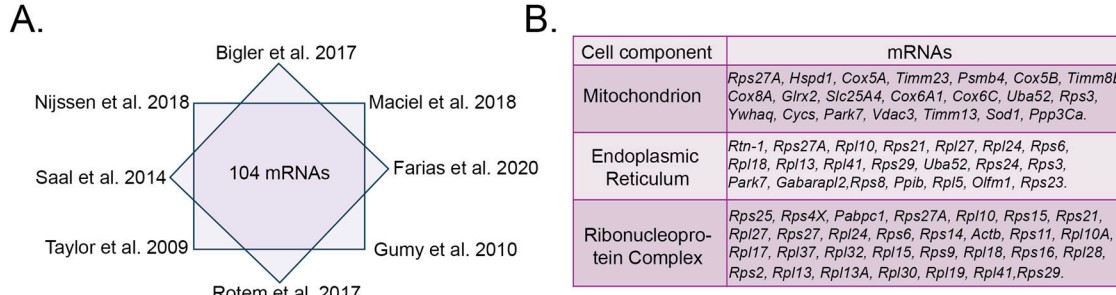

**Figure 1. Analysis of diverse axonal transcriptomes reveals that *Rtn-1* mRNA is consistently identified.**
**(A)** Scheme illustrates studies of axonal transcriptomes and 104 shared mRNAs identified using Venn analysis. **(B)** Table lists the category "Cellular Component" (Gene Ontology, GO) to classify the proteins coded by the 104 common mRNAs. The ER category only harbors *Rtn-1* (bold) as a common mRNA coding for an ER structural protein.

regulating outgrowth. For instance, the physical coupling of the axonal ER to growing microtubules through STIM1 determines its positioning into filopodia in growth cones to control the guidance of developing axons (Pavez et al, 2019). In turn, the axonal ER controls microtubule dynamics, which is required for axonal specification and proper outgrowth in response to injury (Rao et al, 2016; Farías et al, 2019).

Different lines of evidence support that the protein components of the axonal ER may interact with proteins that regulate microtubule dynamics. For instance, the interaction of ER structural proteins with microtubule-severing enzymes may contribute to sculpt the axonal cytoskeleton required for outgrowth (Stone et al, 2012; Rao et al, 2016). As such, the microtubule-severing protein Spastin is known to interact with different ER-shaping proteins (Evans et al, 2006; Mannan et al, 2006), but the contribution of axonal protein synthesis to this process has not been addressed. Here, we hypothesized that the local synthesis of ER structural proteins controls microtubule dynamics and the elongation of injured axons. Using a variety of techniques, we report that

local synthesis of the ER-shaping protein Reticulon-1 (Rtn-1) controls outgrowth and microtubule dynamics of injured cortical axons.

# Results

### *Rtn-1* mRNA is consistently identified in axonal transcriptomes

We hypothesized that the local synthesis of ER structural proteins controls microtubule dynamics and the elongation of injured axons. As a first approach to test our hypothesis, we looked for mRNAs coding for ER structural proteins in reported axonal transcriptomes from different species, developmental stages, and days in vitro (DIV) as indicated in Table 1. To enhance the robustness of our analysis, we focused on transcripts that were consistently identified across conditions using Venn analysis. We identified 104 common mRNAs across all studies (Fig 1A) and classified them under the "Cellular Component" category using

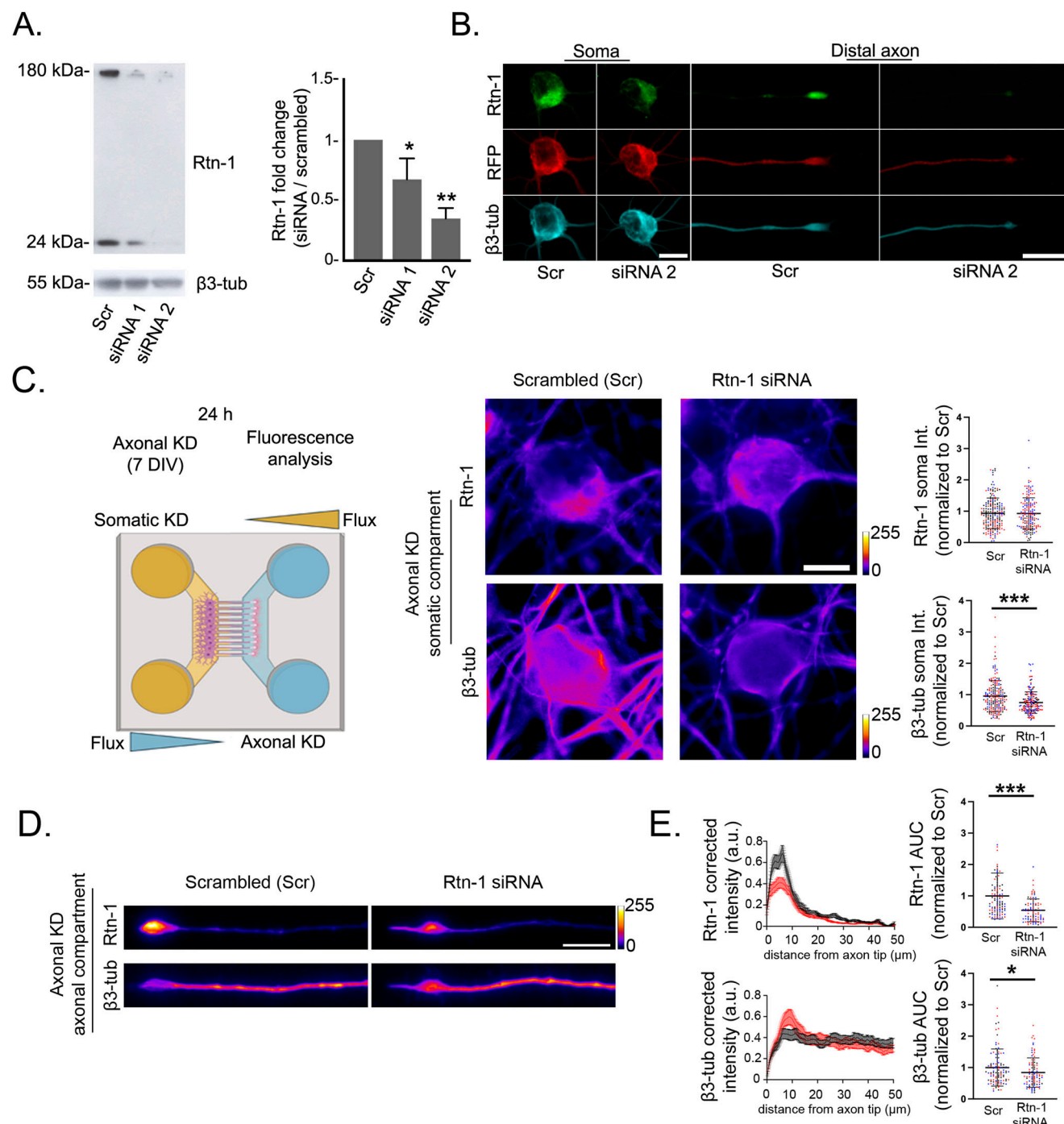

**Figure 2. Axonal synthesis of Rtn-1 can be locally regulated in cortical neurons.**
**(A)** *Left*: Western blot bands of Rtn-1 and β3-tubulin (β3-tub) confirms the knockdown (KD) efficacy of siRNAs specifically designed against *Rtn-1* mRNA (Rtn-1 siRNA). 7 DIV cortical neurons were transfected with Rtn-1 siRNAs and a control sequence (scrambled, Scr) for 48 h. 20 μg of protein was loaded per lane, and β3-tubulin was used as a loading control. *Right*: Graphs quantify the fold change of the 24 kD Rtn-1 band after treatments with siRNAs 1–2 compared with Scr. Data are displayed as the mean ± SE of N = 6 biological replicates. Wilcoxon's test compared with a theoretical mean of 1; *$P < 0.05$; **$P < 0.01$. **(A, B)** Immunofluorescence images of the soma and distal axon of neurons stained against β3-tub and co-transfected with a plasmid encoding RFP and siRNA 2 or Scr under the same conditions indicated in (A). Representative images from N = 4 biological replicates. Bar (soma and axons) = 20 μm. **(C)** *Left*: schematic representation of the axonal KD strategy based on controlling volume differences between the compartment of microfluidic chambers. Under these conditions, 7 DIV cortical neurons were selectively submitted to axonal treatments for 24 h. *Right*: representative images and quantitative analysis of Rtn-1 and β3-tub intensity fold change in the cell bodies after axonal treatments. Bar = 10 μm. Data are displayed as the mean ± SD of 229–300 neurons from N = 3 biological replicates. Linear mixed-effects model with experimental date included as a random effect; ***$P < 0.001$. **(D)** Immunofluorescence showing the intensity changes of Rtn-1 and β3-tubulin along the last 50 μm of axons locally treated with Rtn-1 siRNA or Scr. Bar = 10 μm. **(E)** *Left*: corrected fluorescence intensity profiles of Rtn-1 and β3-tubulin from the last 50 μm of distal axons locally treated with Rtn-1 siRNA (red line) or

Gene Ontology (GO) analysis. We found that *Rtn-1* mRNA was the only transcript coding for an ER structural protein that is present across all the analyzed samples (Fig 1B). Note that several of the identified transcripts in the category "ER" code for cytoplasmic ribosomal components, which indeed can be attached to the axonal ER (Koppers et al, 2024) and be locally synthesized in axons (Shigeoka et al, 2019). On a similar approach, but combining data from dendrites and axons, it was found that *Reticulon-3* mRNA is present in 16 out of 20 studies, further suggesting a wider presence of other mRNAs coding for ER structural proteins in axons (von Kügelgen & Chekulaeva, 2020). Interestingly, one of the studies presented in reference (Fig 1B) showed that *Rtn-1* mRNA levels in cortical axons are decreased by 5.26 ± 0.06-fold after axotomy (Taylor et al, 2009). Thus, *Rtn-1* mRNA is consistently identified in axons, and it is down-regulated upon injury.

Next, we studied the distribution of the Rtn-1 protein in axons from cultured rat cortical neurons at early developmental stages. To this end, we assessed the fluorescence intensity of Rtn-1 along the terminal 50 $\mu$m of distal axons at 1, 3, and 7 DIV. Axonal identity was confirmed by $\beta$3-tubulin and the absence of the dendritic marker microtubule-associated protein 2 (MAP2). Rtn-1 fluorescence was concentrated in the most distal axonal region corresponding to the growth cone (GC), which is significantly more pronounced at 7 DIV compared with 1 DIV (Fig S1A–C). Rtn-1 enrichment within GCs diverged from the staining pattern of KDEL, a luminal ER marker, which did not exhibit such progressive accumulation (Fig S1A–C). The distribution of Rtn-1 agreed with prior studies that localized this protein at the base of the GC's hand-like structure, as outlined by the F-actin cytoskeleton stained with phalloidin (Nozumi et al, 2009) (Fig S1D). This distribution is compatible with Rtn-1 being present in the dense network of ER tubules characteristically found in the GCs, an observation well documented in earlier studies (Deitch & Banker, 1993).

### Axonal synthesis of Rtn-1 lessens outgrowth after injury

To elucidate the role of Rtn-1 synthesis in the outgrowth of injured axons, we first validated knockdown (KD) of this protein by Western blot (WB) in cultured cortical neurons. Neuronal Rtn-1 isoforms include isoform A (Rtn-1A), comprising 776 amino acids, and isoform C (Rtn-1C) with 208 amino acids. Both protein versions are encoded by alternative transcripts (different start sites) of the same gene. Importantly, these isoforms share a highly conserved C-terminal region known as the Reticulon homology domain (RHD), differing only in their 20 N-terminal amino acids (Roebroek et al, 1993; Chiurchiù et al, 2014). Thus, we transfected 7 DIV dissociated cortical neurons with siRNAs designed to target the conserved RHD domain (i.e., siRNAs that target all Rtn-1 isoforms) or a control sequence (scrambled, Scr). Forty-eight hours later, samples were analyzed using a pan-Rtn-1 antibody raised against the conserved RHD region. Two distinct Rtn-1 bands were observed: a higher molecular weight band near 180 kD, likely representing Rtn-1A (Van De Velde et al, 1994), and a lower band near 24 kD, corresponding to

the apparent molecular weight of Rtn-1C (Roebroek et al, 1993; Van De Velde et al, 1994) (Fig 2A). When quantifying the 24-kD band, both siRNAs significantly reduced Rtn-1 levels compared with Scr (Fig 2A). We also examined the 180-kD band and found that siRNA 1 reduced expression to a mean of 0.41 relative to Scr, showing a strong trend that did not reach statistical significance ($P$ = 0.05; N = 3; Wilcoxon's test compared with 1). In contrast, siRNA 2 further reduced expression to a mean of 0.29, which was statistically significant ($P$ = 0.04; N = 3; Wilcoxon's test compared with 1). Together, these results indicate that when considering both isoforms, siRNA 2 achieves the stronger KD. Therefore, unless otherwise indicated, for all subsequent experiments siRNA 2 was used to down-regulate Rtn-1. Immunofluorescence staining further revealed a marked decrease in Rtn-1 intensity in the cell body and distal axons of neurons co-transfected with an RFP-expressing plasmid and siRNA 2 under the same recently described conditions (Fig 2B).

Once we validated the siRNA 2, we aimed at performing KD of Rtn-1 expression selectively within the axonal compartment (axonal Rtn-1 KD) in 7 DIV microfluidic chamber cultures. This compartmentalized approach has been validated in multiple studies to locally control neuronal protein synthesis (Zhang et al, 2013; Baleriola et al, 2014; Batista et al, 2017). Twenty-four hours later, we found that axonal Rtn-1 KD did not alter somatic Rtn-1 levels yet elicited a significant ~25% decrease in somatic $\beta$3-tubulin, which were quantified as the intensity fold change for both proteins compared with Scr (Fig 2C). Compatible with Rtn-1 local synthesis, axonal Rtn-1 KD led to a significant ~46% decrease in its fluorescence intensity within the distal 50 $\mu$m of axons, which was depicted by the corrected fluorescence intensity profile and quantified as the intensity fold change compared with Scr (Fig 2D and E). A slight but significant ~16% decrease in $\beta$3-tubulin fluorescence intensity fold change was also observed after axonal Rtn-1 KD compared with Scr (Fig 2D and E). These results support that local synthesis of Rtn-1 can be regulated in the axonal compartment and suggest an interplay with the organization of the microtubule cytoskeleton.

To investigate the role of local Rtn-1 synthesis in axonal outgrowth, we subjected 8 DIV cortical axons to axotomy by vacuum aspiration in microfluidic chambers. To this end, we simultaneously performed Rtn-1 KD in either axonal or somato-dendritic compartments and measured outgrowth 24 h later (Fig 3A). To measure outgrowth, we took advantage of the semicircle configuration of the NeuroAnatomy FIJI plugin and performed Sholl analysis (Sholl, 1953; McCurdy et al, 2019). We quantified the summation of the intersections of binarized axons with concentric semicircles of 2-$\mu$m increasing radius starting from the microgroove exit and centered at axonal fields of equivalent size (Fig 3B, top). Interestingly, we observed a significant increase in outgrowth (higher number of intersections summation) within the ranges of 500–1,000 $\mu$m and 1,000–1,500 $\mu$m with axonal Rtn-1 KD compared with Scr (Fig 3B, bottom). Furthermore, this response was specific for the axonal compartment, as somatic Rtn-1 KD showed a nonsignificant effect with a slight opposite trend (Fig 3C).

Scr (black line). Intensity profiles depict the mean ± SE from all the analyzed axons. *Right*: bar graphs display the mean ± SD of the fluorescence intensity fold change of 87–102 axons from N = 3 biological replicates. Linear mixed-effects model with experimental date included as a random effect; *$P$ < 0.05; ***$P$ < 0.001.

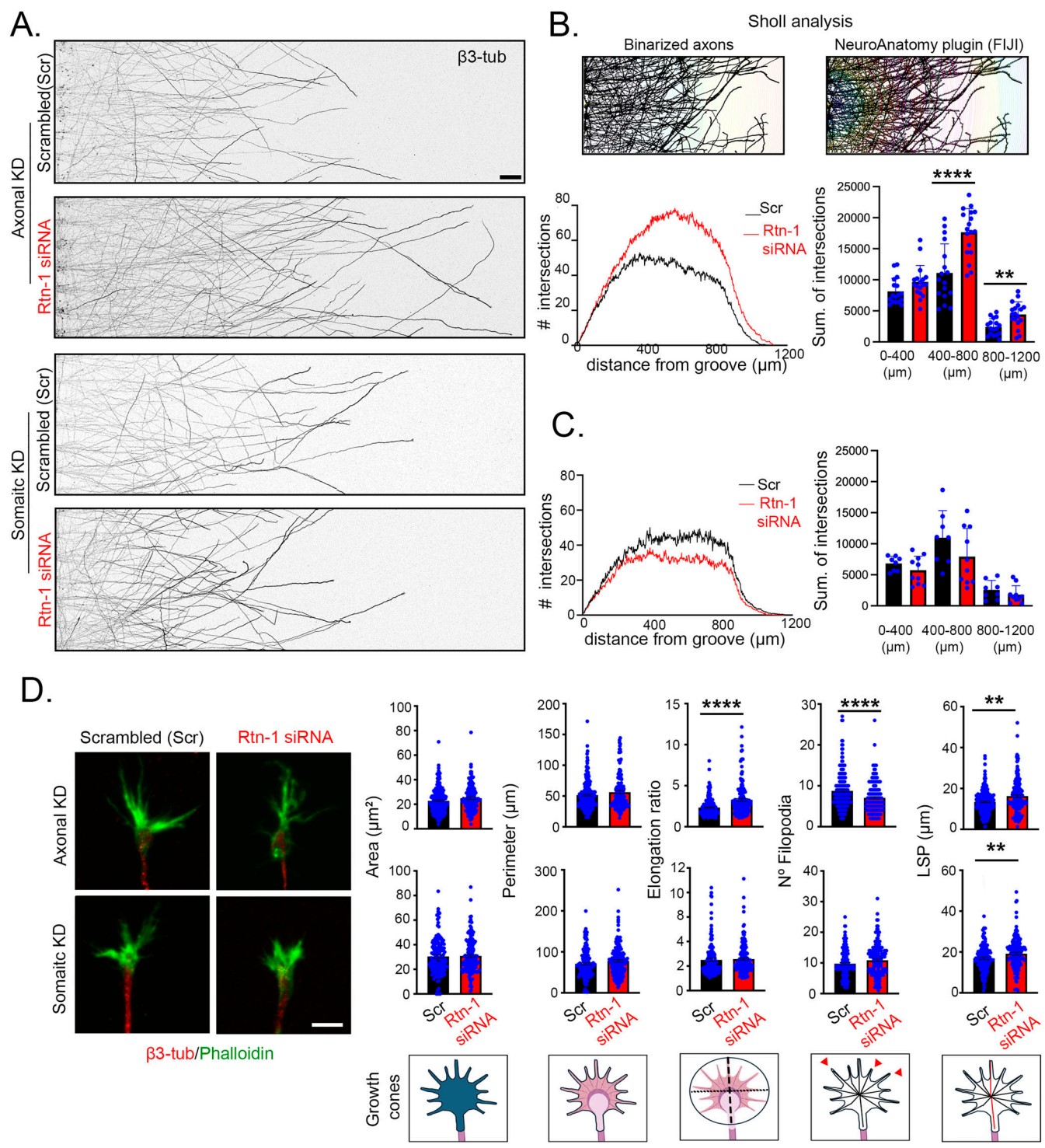

**Figure 3. Axonal Rtn-1 knockdown (KD) promotes outgrowth and modifies growth cone (GC) cytoskeleton after injury.**
**(A)** Immunofluorescence against β3-tubulin in axons growing in microfluidic chambers 24 h after injury (vacuum aspiration axotomy) comparing axonal and somatic Rtn-1 KD performed with a specific siRNA (Rtn-1-siRNA) or control sequence (scrambled, Scr). Bar = 50 μm. **(B)** *Top*: schematic of the Sholl analysis performed with the NeuroAnatomy plugin (FIJI). The number of intersections of binarized axons (*left*) with concentric semicircles of increasing radius starting in the microgroove at the center of each axonal field was quantified (*right*). *Bottom*: mean plots of the Sholl analyses performed after axonal Rtn-1 KD (red line) compared with Scr (black line). Bar graphs show the summation (sum.) of the total intersections within the indicated distance bins. **(C)** Same analysis as in panel B but performed for somatic Rtn-1 KD. Data are displayed as the mean ± SE of 23–24 axonal fields from N = 6 biological replicates for axonal Rtn-1 KD; and 18–20 axonal fields from N = 3 biological replicates for somatic Rtn-1 KD. Two-tailed Mann–Whitney *U* test; **P < 0.01; ****P < 0.0001. **(D)** Images show axonal growth cones after axonal or somatic Rtn-1 KD compared with Scr labeled against β3-tubulin (red) and phalloidin (green) for staining the F-actin outline. Bar = 5 μm. Graphs depict the corresponding differences in GC area, perimeter, elongation ratio (or aspect ratio), number of filopodia, and longest shortest path (LSP), which corresponds to maximal distance calculated from the skeletonized outline

To discard off-target effects of the siRNA used for axonal Rtn-1 KD, we validated a third oligonucleotide sequence (siRNA 3), which was also specifically directed against Rtn-1 conserved domain. Supporting its efficacy, we showed by WB that siRNA 3 transfection of 7 DIV cortical neurons significantly reduced Rtn-1C levels compared with Scr and nontransfected neurons after 24 h. No significant changes of β3-tubulin levels were observed (Fig S2A and B). To test the ability of siRNA 3 to promote outgrowth, we selectively treated axotomized axons in microfluidic chambers and measured outgrowth 24 h later. Axonal KD performed with siRNA 3 resembled siRNA 2 effects (Fig 3A and B). It was observed a significant increase in the summation of axonal intersections within the ranges of 0–500 μm and 500–1,000 μm after axonal Rtn-1 KD compared with Scr (Fig S2C and D). Together, these results suggest that the increased outgrowth of injured axons is not due to siRNA off-target effects.

Next, we assessed the morphology of GCs in the same experimental setting described above. To this end, we used β3-tubulin immune staining and phalloidin incubation to reveal the actin cytoskeleton. Axonal Rtn-1 KD did not modify the outline (perimeter) and the area of GCs. However, axonal Rtn-1 KD produced GCs characterized by a more elongated and compact actin cytoskeleton, reflected on a significant increase in the elongation ratio (calculated as the length of the major axis divided by the length of the minor axis of the best-fitting ellipse to the actin boundary) and longest shortest path (LSP, which reflects the maximum length of the skeletonized actin outline), alongside a reduced number of filopodia (Fig 3D). Collectively, these morphometric parameters revealed that axonal Rtn-1 KD yielded GCs resembling a spade-like morphology that has been previously related to increased outgrowth (Kaethner & Stuermer, 1992). Compatible with this idea, somatic Rtn-1 KD did not promote outgrowth nor modified any of these morphological parameters, except for an increase in the GC LSP (Fig 3D). Together, these results suggest that axonal synthesis of Rtn-1 is normally acting to repress the outgrowth of injured axons.

### Axonal Rtn-1 KD decreases β3-tubulin levels in injured axons

The previously described changes in β3-tubulin elicited by axonal Rtn-1 KD in noninjured axons (Fig 2D) lead us to examine β3-tubulin and Rtn-1 fluorescence intensity after injury. Different from what we observed in intact axons (Fig 2D and E), axotomy concomitant with 24 h of axonal Rtn-1 KD did not alter Rtn-1 fluorescence significantly, but it induced a ~20% reduction in β3-tubulin intensity compared with Scr (Fig 4A). In contrast, somatic Rtn-1 KD did decrease Rtn-1 fluorescence in axons by ~33% (Fig 4B), supporting the notion that most of axonal Rtn-1 protein originates in the soma. Of note, somatic Rtn-1 KD also decreased β3-tubulin intensity by ~19% in axons (Fig 4B).

Suggesting a potential compensatory response of the cell body upon injury, axonal Rtn-1 KD leads to increased somatic Rtn-1 fluorescence compared with Scr (Fig 4C). Interestingly, an isoform-specific increase in Rtn-1C levels has been previously described upon the injury of cortical neurons (Fan et al, 2018). Although we still do not know the mechanism, the somatic increase that we observe may partially explain the lack of changes in axon Rtn-1 levels after 24 h of axonal KD (Fig 4A), which is potentially countered by somatic Rtn-1 synthesis. Intriguingly, resembling the somatic increase of Rtn-1, β3-tubulin levels were also elevated in the cell bodies after axonal Rtn-1 KD compared with Scr (Fig 4C). In contrast, somatic Rtn-1 KD resulted in a significant reduction in the somatic fluorescence intensity for both proteins compared with Scr (Fig 4D). Together, these results indicate that after injury, the levels of Rtn-1 in the somatic and axonal compartments tend to change in a manner that parallels β3-tubulin.

### Spastin activity regulates β3-tubulin and axon regrowth under local Rtn-1 KD

The significant alterations in β3-tubulin levels after Rtn-1 KD in the context of injury led us to explore the role of Rtn-1 interactors that can modify the microtubule cytoskeleton. Notably, a previous study had shown by yeast two-hybrid assay that a protein sequence from Rtn-1C directly interacts with the microtubule-severing protein Spastin (Mannan et al, 2006). Therefore, we explored the localization of Spastin and Rtn-1C in distal axons before and after injury.

To this end, we stained 9 DIV noninjured and injured axons (submitted to axotomy at 8 DIV) against Spastin (all isoforms) and Rtn-1C to perform stimulated emission depletion (STED) super-resolution microscopy. In both conditions, Rtn-1C and Spastin showed enrichment of the staining in the GC (Fig 5A, left panel). The distribution of Rtn-1C along the axonal shaft is fine and continuous, consistent with its localization in ER tubules, which appear decorated with clusters of Spastin. In contrast to the more central positioning of Rtn-1C in the axonal shaft, Spastin displays a broader distribution in both conditions (Fig 5A, left panel). This difference is also seen in the GCs, where Rtn-1C is more concentrated at the center of these domains (Fig 5A, left panel). In injured axons, distal Rtn-1C staining appears less compact, adopting a more tubulated pattern (Fig 5A, left panel). Moreover, after injury, Rtn-1C staining seems slightly retracted from the GC compared with the noninjured condition, as also suggested by its relative intensity profile (Fig 5A, right panel). Spastin distribution along the distal axon remains similar between conditions (Fig 5A, right panel). However, in injured axons, Spastin appears less diffuse and more concentrated in clusters than in noninjured axons (Fig 5A, left and right panel). No significant changes were observed when analyzing the total fluorescence intensity of both proteins (Fig S3). Although we did not directly test their direct molecular association, these results are consistent with Rtn-1C and Spastin sharing a similar subcellular localization, potentially enabling their functional interaction in distal axons.

Next, we explored whether axonal Rtn-1 KD is related to Spastin's microtubule severing after injury. We subjected 8 DIV

(shown with the red line of the schematic drawings below). Data are displayed as the mean ± SE of 145–153 axons for axonal KD; 227–148 axons for somatic KD from N = 3 biological replicates. Two-tailed Mann–Whitney U test; **P < 0.01; ****P < 0.0001.

Rtn-1 synthesis controls outgrowth in injured cortical axons   Luarte et al.    https://doi.org/10.26508/lsa.202503571   vol 9 | no 4 | e202503571   **6 of 23**

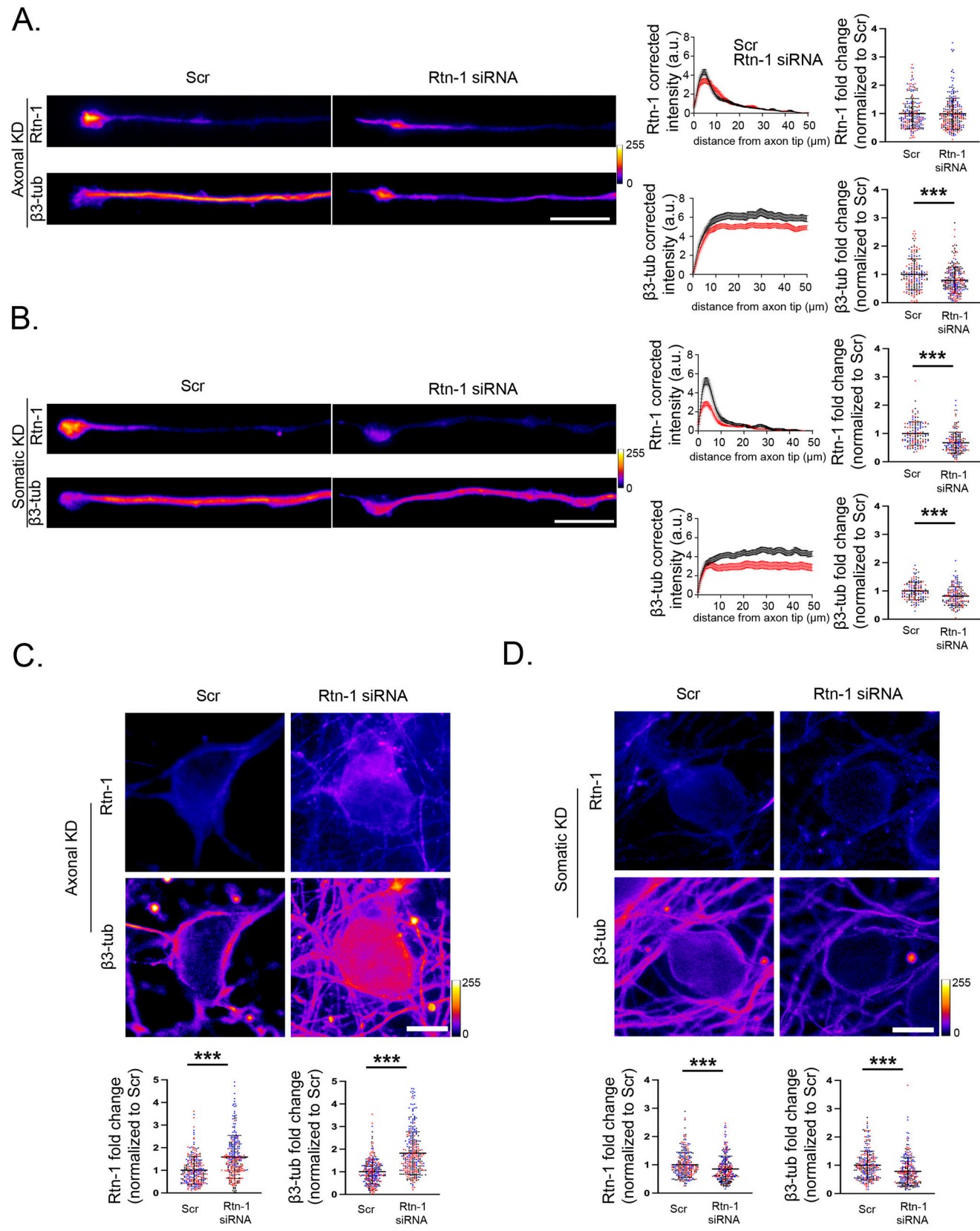

neurons to axotomy and axonal treatments, but this time we analyzed them in a twofold longer period (48 h) compared with the previous experiments shown in Figs 3 and 4. This extended time period was taken to (i) reveal more clearly the previously observed changes in β3-tubulin content and (ii) allow the proper expression of the EB3-GFP used for the live-cell imaging described later in Fig 5. For these experiments, we took advantage of Spastazoline (SPTZ), a well-characterized and specific inhibitor of the ATPase activity of Spastin (i.e., microtubule severing) (Cupido et al, 2019). Resembling the results of Fig 4, we found that axonal Rtn-1 KD resulted in a ~60% reduction in β3-tubulin levels within the distal 50 μm of axons, compared with controls (Scr + DMSO) (Fig 5B). However, co-incubation with 50 nM SPTZ (applied 4 h after axotomy) completely abolished the β3-tubulin reduction elicited by axonal Rtn-1 KD, whereas it did not alter Rtn-1C and Spastin levels (Fig 5B). This rescue of β3-tubulin levels during axonal Rtn-1 KD by a specific drug (SPTZ) makes it unlikely an off-target effect of the siRNA over β3-tubulin mRNA. If this was the case, this potential off-target effect should be still operating in the presence of SPTZ. Also discarding unspecific effects over other proteins, SPTZ administration alone did not significantly modify β3-tubulin levels, nor those of Rtn-1C and Spastin (Fig 5B). Together, these observations suggest that axonal Rtn-1 KD correlates with higher Spastin microtubule severing.

Next, we studied whether the effects of axonal Rtn-1 KD on outgrowth depended on Spastin's microtubule severing. In contrast to the effects observed at 24 h from axotomy (Fig 3A), after 48 h of axonal Rtn-1 KD there was only a trend (nonstatistically significant) to increase outgrowth in the range of 1,000–1,500 μm compared with controls (Fig 5C and D). However, local co-incubation of 50 nM SPTZ with axonal Rtn-1 KD abolished this trend, producing a statistically significant decrease in outgrowth compared with the axonal Rtn-1 KD + DMSO condition (Fig 5C and D). Local SPTZ application in axons treated with a Scr sequence displayed a trend to increase outgrowth compared with all the conditions, particularly in the 1,000- to 1,500-μm segment, but none of these comparisons reached statistical significance (Fig 5C and D). Together, these results suggest a relationship between axonal Rtn-1 KD and Spastin microtubule severing in modulating outgrowth.

Considering the observed shifts in microtubule content and the previously reported role of Spastin in controlling microtubule stability (Teixeira Lopes, 2018; Kuo et al, 2019; Aiken & Holzbaur, 2024 Preprint), we aimed at quantifying microtubule dynamics by live-cell imaging before and after injury. To this end, cortical neurons cultured in microfluidic chambers were transfected at

7 DIV with a plasmid coding for the microtubule tip-binding protein EB3 fused to the fluorescent construct Emerald GFP (EB3-GFP), which labels the growing plus ends of polymerizing microtubules (Stepanova et al, 2003). Next, axotomy was performed at 8 DIV microfluidic chambers, and immediately, axons were locally transfected with an Rtn-1 siRNA or Scr sequence. Noninjured 8 DIV axons were transfected in parallel. Four hours after transfection, noninjured and injured axons were treated with either DMSO (vehicle) or 50 nM SPTZ. Then, 48 h since injury (at 10 DIV), axons were directly imaged in microfluidic chambers to examine the dynamic behavior of EB3-GFP puncta (comets).

We first analyzed the effect of axonal Rtn-1 KD in noninjured axons. Interestingly, we found that axonal Rtn-1 KD elicited a statistically significant increase in microtubule growth rate, whereas it did not significantly modify microtubule track length and lifetime compared with controls (Scr + DMSO) (Fig 5E). To reveal the role of Spastin's microtubule severing in modifying these parameters, we co-incubated axons with 50 nM SPTZ. As such, SPTZ treatment did not prevent the increase in microtubule growth rate elicited by axonal Rtn-1 KD nor modified comet's lifetime but induced a slight increase in track length when compared to controls (Fig 5E).

Next, we analyzed the effect of axonal Rtn-1 KD on comet parameters after injury. In contrast to the milder increase observed in noninjured axons, axonal Rtn-1 KD augmented by nearly twofold the growth rate and track length of microtubules, whereas it did not modify comet lifetime compared with controls (Fig 5E, left and right panels). Of note, compared with noninjured axons, growth rate and track length are significantly decreased on axotomized axons (from 0.09 to 0.06 μm/s, and from 4.6 to 2.6 μm; P < 0.0001, Mann–Whitney test). These results are compatible with axonal damage promoting a more unorganized filament or less processivity in the microtubule growth (Yap et al, 2017; Rolls et al, 2020). SPTZ co-treatment did not prevent the increase in microtubule growth rate and track length elicited by axonal Rtn-1 KD after injury compared with controls, nor modified comet's lifetime (Fig 5E, right panels). However, SPTZ treatment along with axonal Rtn-1 KD did induce a slight increase in comet's lifetime when compared to axonal Rtn-1 KD + DMSO as shown in right panels of Fig 5E. Together, these results suggest that axonal Rtn-1 synthesis controls microtubule dynamics in both noninjured and injured axons, independently of Spastin-mediated microtubule severing.

Collectively, results are compatible with Rtn-1 synthesis counteracting Spastin microtubule severing for decreasing outgrowth after injury. Previous evidence points to a specific role of the

**Figure 4. Axonal and somatic β3-tubulin levels are dependent on the local synthesis of Rtn-1 after injury.**
**(A)** *Left*: immunofluorescence images and their corresponding corrected fluorescence intensity profiles along the last 50 μm of distal axons stained against Rtn-1 and β3-tubulin after axonal transfection with Rtn-1 siRNA (red; axonal Rtn-1 KD) or a control sequence (black; scrambled, Scr). Intensity profiles depict the mean ± SD from all the analyzed axons. Bar = 10 μm. *Right*: bar graphs quantify the Rtn-1 and β3-tubulin (β3-tub) fluorescence intensity fold change. Data are displayed as the mean ± SD of 206–242 axons from N = 3 biological replicates. Linear mixed-effects model with experimental date included as a random effect; ***P < 0.001. **(B)** Same analysis as in panel A but performed for somatic Rtn-1 KD. Data are displayed as the mean ± SE of 133–161 axons from N = 3 biological replicates. Linear mixed-effects model with experimental date included as a random effect; ***P < 0.001. **(C)** Images depict neuronal cell bodies stained for Rtn-1 and β3-tub after axonal Rtn-1 KD or Scr. Bar graphs depict Rtn-1 and β3-tub fluorescence intensity fold change displayed as the mean ± SE of 358–382 neurons from N = 3 biological replicates. Linear mixed-effects model with experimental date included as a random effect; ***P < 0.001. Bar = 10 μm. **(C, D)** Same analysis as in panel (C) but performed for somatic Rtn-1 KD. Data are displayed as the mean ± SE of 252–237 neurons from N = 3 biological replicates. Linear mixed-effects model with experimental date included as a random effect; ***P < 0.001. Bar = 10 μm.

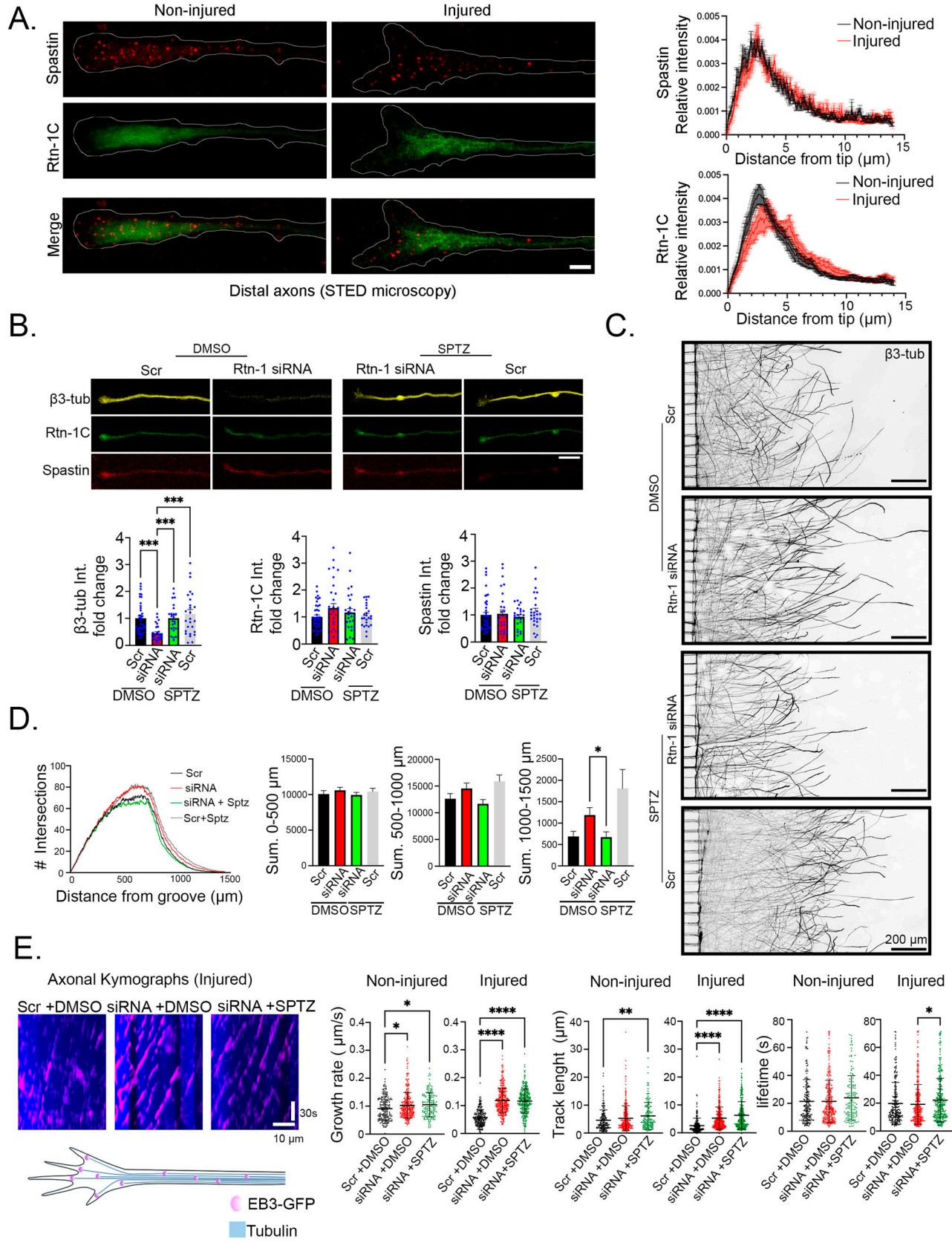

Rtn-1C isoform in this potential interplay (Mannan et al, 2006). However, it is still unknown which Rtn-1 isoform is locally produced in axons.

### The Reticulon-1 isoform C (Rtn-1C) is locally synthesized in cortical axons

We sought to characterize the isoform expression of *Rtn-1* mRNA and protein in both axons and cell bodies. Because microfluidic chambers yield only limited cellular material, we adopted an alternative culture approach using "neuronballs." This method enables the segregation of an axon-enriched fraction by mechanically separating axons from somato-dendritic structures. The axonal compartment of desomatized neuronballs was identified by IF against medium chain neurofilament (NF-M), whereas the somato-dendritic compartment was stained with microtubule-associated protein 2 (MAP2) (Fig 6A). RNA was then extracted from both the axonal and somato-dendritic fractions and submitted to reverse transcription PCR (RT–PCR). The purity of the axonal fraction was evidenced by the absence of amplification for *Map2*, *Histone h1* (general nuclei marker), and the astrocyte-specific glial fibrillary acidic protein (*Gfap*) mRNAs. In contrast, we found positive amplification for *β-actin* transcript, which is known to be present in axonal compartment (Fig 6A). These fractions were further analyzed with *Rtn-1*–specific primers targeted against C-terminal domain or the N-terminal domain (to reveal *Rtn-1A*). In contrast to the somato-dendritic fraction, axons did not amplify for *Rtn-1A* mRNA, suggesting that *Rtn-1C* transcript is the predominant isoform in this compartment (Fig 6A).

Next, we investigated the protein composition of the axonal and somato-dendritic fractions of neuronballs by WB. Consistent with the sample purity observed for mRNAs, the axonal fraction was positive for Tau and β3-tubulin, with no detection of the dendritic marker MAP 2 (Fig 6A). For Rtn-1 identification, we employed a pan-Reticulon-1 antibody targeting the conserved C-terminal Reticulon homology domain (RHD). Importantly, the axonal fraction showed exclusivity for the lower molecular weight band, indicative of Rtn-1C presence, mirroring RT–PCR results (Fig 6A). Immunofluorescence analysis of Rtn-1C in axons from 8 DIV microfluidic chambers displayed a distribution pattern consistent with that of the pan-Reticulon-1 (Rtn-1 antibody), as illustrated in Fig 2 (Fig 6B). In contrast, Rtn-1A presence was limited to a small subset of axons,

exhibiting a peak of localization near the growth cone base (Fig 6B). Thus, resembling the distribution of transcripts, Rtn-1C protein appears as the predominant isoform in the axonal compartment.

In Fig 2, we provided evidence compatible with the axonal synthesis of Rtn-1 in cortical axons. In addition, we showed that the predominant mRNA and protein in cortical axons correspond to the Rtn-1C isoform (Fig 6A and B). However, whether Rtn-1C is indeed locally synthesized in cortical axons has not been proved.

To demonstrate the local production of Rtn-1C in axons, we employed a proximity ligation assay integrated with puromycylation (puro-PLA) (Dieck et al, 2015). During this assay, we selectively administered a puromycin pulse (15 min) to the axonal or somato-dendritic compartment of microfluidic chambers, allowing its covalent incorporation into nascent polypeptides. This enabled the detection of newly synthesized Rtn-1C peptides through fluorescent signals produced by the nanometric proximity (<40 nm distance) of puromycin and Rtn-1C antibodies (Fig 6C). Positive amplification was observed in the somato-dendritic compartment of microfluidic chambers with local puromycin treatment as dots or puncta (Fig 6D, left panel). However, virtually no signal was detected in the cell bodies when puromycin was added to the axonal compartment, showing the effective compartmentalization of this approach (Fig 6D, left panel). Positive signal in the cell bodies was abrogated when the protein synthesis inhibitor emetine was previously added, further supporting that these signals reflect protein synthesis (Fig 6D, left panel). Supporting its axonal synthesis, fluorescent puncta were observed in axons, which was significantly reduced by previous emetine treatment (Fig 6D, right panel). Interestingly, when the puromycin pulse was applied exclusively to the somatic compartment, positive signal was still detected in axons in a significantly lower amount compared with direct and puromycin treatment, but still significantly higher than the emetine/puromycin condition (Fig 6D, right panel). This suggests that at least a fraction of puromycylated peptides synthesized in the cell bodies may rapidly (within 15 min) reach the axons through as-yet-uncharacterized mechanisms. Nevertheless, the significantly stronger signal detected in the axonal compartment with direct puromycin treatment is fully consistent with local Rtn-1C synthesis in axons.

To further validate puro-PLA detection, we tested the Rtn-1C antibody specificity by WB and IF, and whether puro-PLA signals depended on local depletion of *Rtn-1* mRNA (axonal Rtn-1 KD). Because of their low background for neuronal components, we used COS-7 cells expressing the rat Rtn-1C protein fused to GFP (Rtn-1C-

**Figure 5. Axonal Rtn-1 KD after injury controls distal tubulin levels and outgrowth depending on Spastin microtubule severing.**
**(A)** *Left*: representative fluorescent super-resolution STED images of distal noninjured and injured cortical axons (44 h post-axotomy) in microfluidic chambers, which were stained against Rtn-1C (green) and Spastin (red); axon outlines are depicted (white lines). *Right*: plots depict the relative fluorescence intensity profile (normalized against total distal intensity) of Rtn-1C and Spastin in noninjured (black) and injured (red) axons. Data are displayed as the mean ± SE of 9–13 axons from N = 4 biological replicates. Bar = 5 µm. **(B)** *Top*: representative confocal images of distal axons stained against β3-tubulin (β3-tub, yellow), Rtn-1C (green), and Spastin (red) in microfluidic chambers 44 h post-axotomy under indicated treatments: control (Scrambled, Scr), axonal Rtn-1 KD (Rtn-1 siRNA), and Spastin pharmacological inhibition with 50 nM Spastazoline (SPZT) or vehicle (DMSO). Bar = 10 µm. *Bottom*: quantifications of fluorescence intensity fold change of indicated proteins in the last 50 µm normalized to Scr + DMSO. Data are displayed as the mean ± SE of 24–40 axons for condition from N = 3 biological replicates. Kruskal–Wallis followed by Dunn's post hoc test; ***P < 0.001. **(B, C)** Illustrative axonal fields in microfluidic chambers 44 h post-axotomy stained against β3-tub under the treatments shown in (B). Bar = 200 µm. **(D)** *Left*: mean plots of the Sholl analyses performed with colored lines indicating the analyzed conditions. *Right*: bar graphs show the summation (Sum.) of the total intersections within the indicated distance bins. Data are displayed as the mean ± SE of 22–24 axonal fields for condition from N = 3 biological replicates. One-way ANOVA followed by Tukey's post hoc test; *P < 0.05. **(E)** *Left*: axonal kymographs showing microtubule dynamics in neurons with indicated treatments 44 h after axotomy; dashes represent the traces of EB3-GFP (end-binding protein 3 fused to Emerald GFP). Scale bars represent 10 µm and 30 s, respectively. *Right*: bar graphs show microtubule dynamic parameters in noninjured and injured axons with indicated treatments. Data are displayed as the mean ± SE of 146–377 comets/dashes for condition from N = 3 biological replicates. Kruskal–Wallis followed by Dunn's post hoc test; *P < 0.05; **P < 0.01; ****P < 0.001.

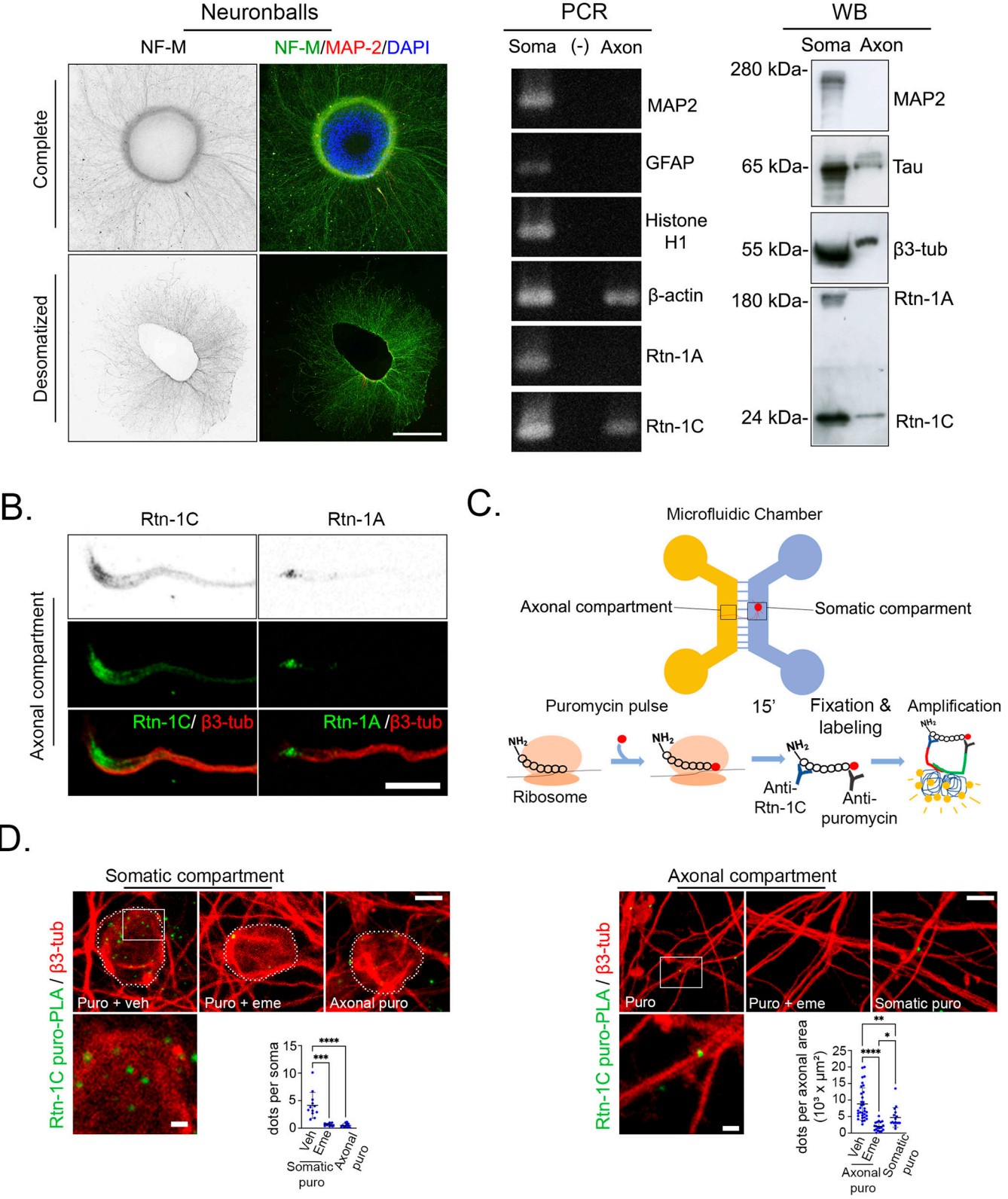

GFP) and analyzed them by WB, using the same Rtn-1C antibody of the puro-PLA experiment. As expected, a single band of the predicted molecular weight for the recombinant protein was observed in the lane of transfected cells, whereas no band was apparent in nontransfected cells (Fig S4A). Also, revealing an isoform-specific immunodetection, the ER pattern depicted by Rtn-1C-GFP fully overlapped with the IF using the Rtn-1C antibody (Fig S4B). Similar results were observed for Rtn-1A-GFP–expressing COS-7 cells stained against Rtn-1A antibody (Fig S4B). Furthermore, we tested the specificity of this antibody by transfecting cells with siRNA 2 sequence. As expected, siRNA 2 induced a significant decrease in Rtn-1C band compared with Scr (Fig S4C). We also validated the presence of *Rtn-1* mRNA in axons cultured under similar conditions (9 DIV microfluidic chamber) as shown by Fig S4D. Finally, 9 DIV axons were locally treated in microfluidic chambers with siRNA 2 for 24 h, which produced a significant decrease in positive puro-PLA signal compared with Scr (Fig S4D, right). Together, these results strongly suggest that the quantified puro-PLA signal is reflecting the local translation of *Rtn-1C* mRNA in axons.

### Rtn-1C but not Rtn-1A reduces the microtubule-severing activity of Spastin

We described that Rtn-1C is locally synthesized in cortical axons, whereas Rtn-1A synthesis should predominantly occur in the cell body. Therefore, we sought to understand the functional implications of such compartmentalization. We chose COS-7 cells as a heterologous system because of their increased co-transfection efficiency, flat and extended morphology, and low background of neuronal proteins. These features provided an advantageous context to visualize ER protein distribution and functional relationships.

Thus, we first overexpressed in COS-7 cells an RFP-tagged version of the most abundant Spastin isoform during neuronal development, which corresponds to M87 isoform. Of note, in contrast to M1 isoform, M87 isoform is expected to be soluble and devoid of the ER-shaping membrane domain (Solowska et al, 2017). Next, the plasmid coding for isoform M87 was co-transfected with Rtn-1C-GFP, Rtn-1A-GFP, or GFP alone. Spastin-RFP displayed a punctate distribution when co-expressed with GFP, indicative of its potential interaction with endosomal components, as reported elsewhere (Allison et al, 2013) (Fig 7A, upper panels). Under this condition, Spastin puncta strongly co-distributed with some clusters of soluble GFP. It has been described that under certain

circumstances, soluble proteins such as GFP might be bulk up taken by late endosomal structures (Sahu et al, 2011), which are also known to be positive for M87 Spastin (Allison et al, 2019). To rule out that, in our case, this co-distribution is due to fluorescence crosstalk, we independently acquired images for each fluorophore in co-transfected cells. We observed that COS-7 cells co-transfected with Spastin-RFP and GFP displayed the same previously described puncta for both channels, discarding crosstalk (Fig S5). Further discarding unspecific channel bleeding, some cells that exclusively express Spastin-RFP harbor puncta that are exclusively seen in one channel, as indicated in Fig S5. Similarly, some strong GFP puncta (indicated in the image) are not detected in the RFP channel (Fig S5). Complementarily, cells co-transfected with Rtn-1A-RFP and GFP show distinct distribution patterns without overlapping puncta, suggesting that the punctate distribution of GFP is associated exclusively with Spastin co-expression and not explained by unspecific bleeding (Fig S5).

Of note, Spastin-RFP pattern was drastically altered when co-expressed with Rtn-1C-GFP (Fig 7B, middle panels). As such, and potentially compatible with their previously reported direct interaction, Spastin closely followed the Rtn-1C pattern distribution (Fig 7B, middle panels). Conversely, Rtn-1A-GFP showed minimal overlap with Spastin-RFP (Fig 7B, lower panels). Quantification of protein colocalization by Manders' coefficients substantiated these observations (Fig 7C).

Notably, the co-expression of Spastin-RFP with Rtn-1C-GFP maintained tubulin levels unaltered when compared to non-transfected cells from each experimental condition, in marked contrast to their GFP and Rtn-1A-GFP counterparts (Fig 7B and C). Furthermore, the filamentous distribution of tubulin was clearly preserved with Spastin-RFP/Rtn-1C-GFP co-expression, which is a strong indicative of limited severing of microtubule filaments (Fig 7B, middle panels). These effects cannot be attributed to differences in transfection efficiency, as fluorescence intensity of Spastin-RFP and GFP constructs did not significantly differ across conditions (Fig S6). Therefore, these results suggest that the tight co-distribution of Rtn-1C with Spastin is associated with reduced microtubule-severing activity, which is not observed for the Rtn-1A isoform.

The examination of the sequences of Rtn-1C and 1A reveals a unique N-terminal variance as the point of differentiation between both isoforms, as they share their RHD region (Fig 7D). Considering the limited colocalization of Rtn-1A with Spastin (Fig 7B, lower

---

**Figure 6. Rtn-1C is locally synthesized in cortical axons.**
**(A)** *Left*: immunofluorescence of neuronballs stained against neurofilament medium chain (NF-M, grayscale and green) and microtubule-associated protein 2 (MAP2, grayscale and red) with nucleus counterstaining (DAPI, blue). The top panel shows intact neuronballs, and bottom panel shows them after mechanic removal of the cell bodies. *Middle*: RT–PCR bands of somatic (soma), axonal (axon), and (–) (reaction of axonal RNA without reverse transcriptase) to reveal mRNAs coding for MAP2, glial fibrillary acidic protein (GFAP), Histone H1, β-actin, Rtn-1C, and Rtn-1A. *Right*: Western blot analysis of the corresponding fractions using a pan-Rtn-1 antibody to reveal Rtn-1A and Rtn-1C isoforms, β3-tubulin (β3-tub), and Tau. Bar = 500 μm. Images and bands are representative of two biological replicates. **(B)** Confocal images showing the expression of Rtn-1C and Rtn-1A (green) and β3-tub (red) in the distal axonal compartment of microfluidic chambers. Bar = 10 μm. Images are representative of two biological replicates. **(C)** Schematic of the microfluidic chamber setup used to perform proximity ligation assay coupled to puromycylation (puro-PLA) is depicted below. Local puromycin (puro) incorporation into newly synthesized peptides allows amplification when puro is at nanometric distance from the N terminus. **(D)** puro-PLA–positive spots are shown in green at somatic and axonal compartments. β3-tub labeling depicts cell bodies and axons. Insets show higher magnification of puro-PLA signals. Bar graphs quantify dots per cell bodies and per axonal area after local co-treatments of puro with the protein synthesis inhibitor emetine (Eme) or vehicle (Veh), or puro at the opposite compartment. Bar = 5 μm; inserts = 1 μm. Data are displayed as the mean ± SD of 11–12 somatic and 16–33 axonal fields from N = 4 biological replicates. Kruskal–Wallis followed by Dunn's post hoc test; *P < 0.05; **P < 0.01; ***P < 0.001; ****P < 0.0001.

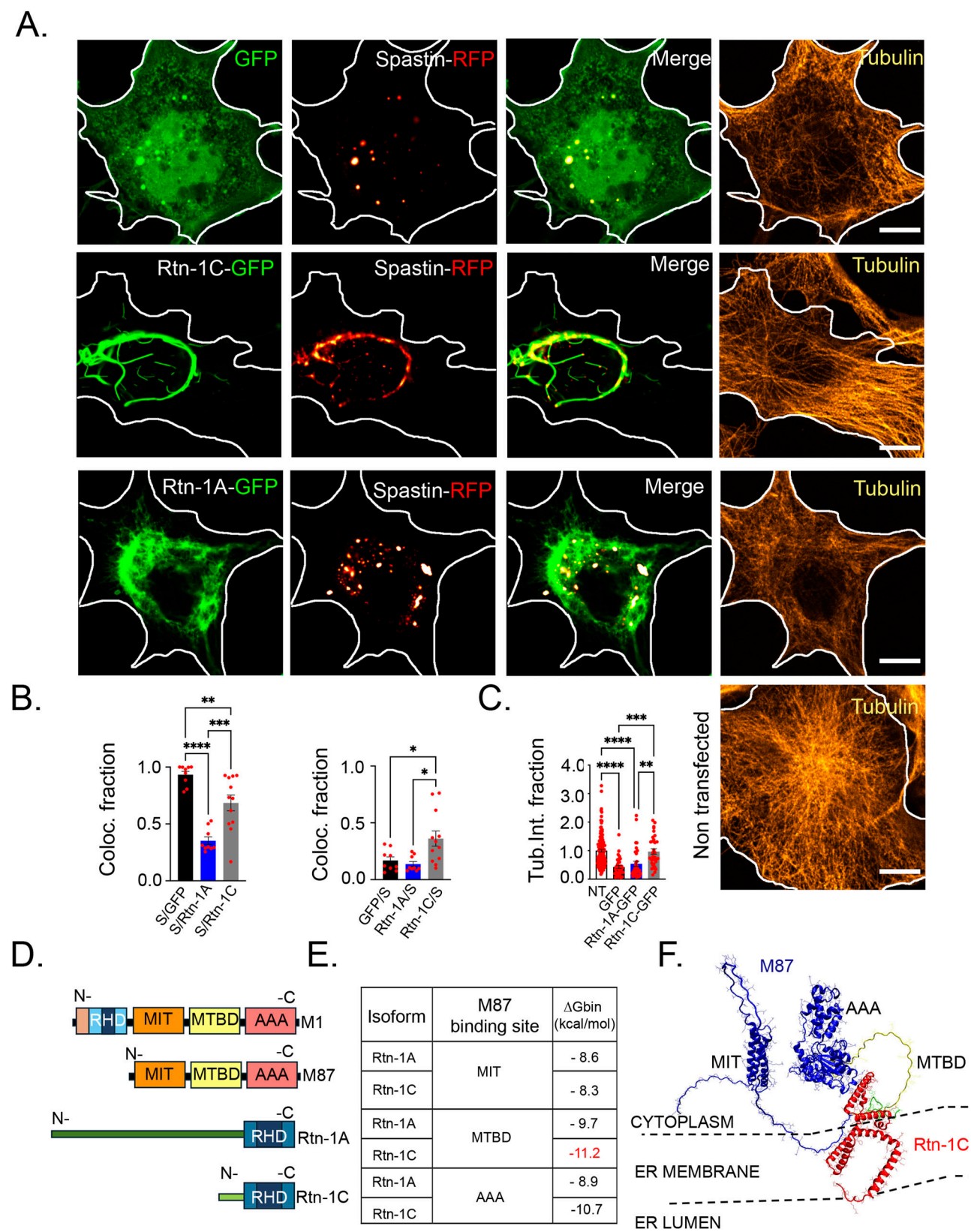

**Figure 7. Rtn-1C co-distributes with Spastin and attenuates its microtubule-severing activity.**
**(A)** Representative images of COS-7 cells expressing Spastin-RFP (fire red) together with GFP (green), Rtn-1A-GFP, or Rtn-1C-GFP, and immunostained against tubulin (yellow). Bar = 10 μm. **(B)** Bar graphs show indicated Manders' coefficients to quantify the colocalization between Spastin-RFP and GFP or Rtn-1 isoforms. Data are

panels), RHD appears not to be sufficient for their interaction. This observation hints at the exclusive 20 N-terminal amino acids of Rtn-1C as being critical for Spastin binding. Therefore, we performed in silico modeling of the Rtn-1C N-terminal and anticipated its affinity with the microtubule-binding domain (MTBD) of Spastin. This domain showed larger interaction energies compared with the endosome (Microtubule Interacting and Trafficking domain, MIT) and microtubule-severing domain (ATPases Associated with diverse cellular Activities, AAA) (Fig 7E). Furthermore, the affinity of a Rtn-1A equivalent region for these domains was also less thermodynamically favored compared with Rtn-1C (Fig 7E). These results are compatible with previous evidence showing that deletion of Spastin's MIT domain, and most of the MTBD region, abrogates Spastin-Rtn-1C co-precipitation (Mannan et al, 2006). This suggests that Rtn-1C binds to Spastin MTBD as indicated in Fig 7F, competing with the binding of this domain to microtubules. Mechanistically, this would prevent the ring conformation around microtubule filaments required for the severing activity of Spastin (Roll-Mecak & Vale, 2008). Overall, these findings suggest that Rtn-1C, but not Rtn-1A, may associate with Spastin to reduce its microtubule-severing activity.

# Discussion

This study suggests a mechanism whereby down-regulating Rtn-1 local synthesis promotes the outgrowth of injured cortical axons and microtubule dynamics. More specifically, our results are compatible with outgrowth dependent on a potential shielding effect that Rtn-1C may exert over Spastin microtubule severing. This injury adaptation is intriguing given that its halting effect on outgrowth may seem, at first sight, detrimental for proper regeneration. However, it is possible that limiting microtubule turnover at injury site helps axons to accumulate molecular components required for a successful injury repair. Whether the control of local synthesis of Rtn-1 offers a potential therapeutic pathway to enhance axonal regeneration by unleashing its shielding effect over Spastin remains to be further explored.

### Control of Rtn-1 synthesis as an adaptive mechanism for axonal injury

It would also be interesting to further explore whether developing neurons decrease axonal *Rtn-1* mRNA translation as a mechanism to promote outgrowth during development and after damage. For instance, a previous study using axotomized 10 DIV rat cortical neurons in microfluidic chambers found a decrease in *Rtn-1* mRNA levels compared with noninjured axons (Taylor et al, 2009). The similarity between phenotypes that we

reported before and after injury, such as decrease in distal *β*3-tubulin levels and increased microtubule growth rate, is compatible with Rtn-1C synthesis occurring before and after injury. However, a more definite answer might be provided by performing puro-PLA measurements of Rtn-1C synthesis before and after injury, which is an ongoing work.

Currently, it is unknown to what extent a similar mechanism is retained in adult cortical neurons. Likewise, we do not know whether this phenomenon is fully operative in vivo, even though *Rtn-1* mRNA is consistently found in vitro and ex vivo. Of note, *Rtn-1* mRNA is dynamically regulated in the axonal translatome of retinal ganglion cells in vivo (Shigeoka et al, 2016), but the functional implications of axonal Rtn-1 synthesis in a more physiological context are yet to be elucidated. Interestingly, growing axonal tips from injured nerves in Lamprey are nearly 13-fold enriched in *Spastin* mRNA, suggesting that other constituent of our described mechanism may be locally synthesized after injury (Jin et al, 2022).

The presence of the axonal ER (particularly the smooth ER) is thought to be critical for promoting the regeneration of injured neurites by providing membranes and controlling calcium and microtubule dynamics required for outgrowth (Luarte et al, 2018). This has been supported by evidence showing the distribution of smooth ER markers, including ER-shaping proteins such as Reticulons, REEPs, Atlastin, and Spastin isoforms, preferentially at the distal tip of regenerating neurites (Stone et al, 2012; Rao et al, 2016). Indeed, Reticulons share RHD with REEPs, some Spastin isoforms, and Atlastins, which makes them suitable to shape the mostly tubular smooth ER (Iwahashi et al, 2007; Chiurchiù et al, 2014). Highlighting their functional relevance, forcing the localization of smooth ER into axonal tips by the overexpression of the ER membrane protein Protrudin enhances outgrowth of CNS axons (Petrova et al, 2020).

Different studies have shown that the presence of the axonal ER is associated with the fine-tuning of microtubules stability to sustain axonal extension (Lee et al, 2009; Farías et al, 2019). Interestingly, our results are consistent with axonal Rtn-1 synthesis controlling microtubule growth rate in elongating axons.

Some studies have pointed to a noncanonical role of Rtn-1C synthesis by signaling in the nucleus, potentially through DNA binding and histone deacetylase inhibition (Nepravishta et al, 2010, 2012). It is tempting to speculate that these still emerging roles may also contribute to the observed phenotypes. Of note, different axonally synthesized proteins exert transcriptional control after in response to injury or local cues (Twiss et al, 2016).

The differential injury response in axonal outgrowth observed after somatic or axonal Rtn-1 KD emphasizes the unique molecular and structural environments of axons versus cell bodies. This difference might be explained by at least two scenarios. First, we

---

displayed as the mean ± SE from nine cells per condition from N = 2 biological replicates. One-way ANOVA with Tukey's post hoc test; *$P < 0.05$; **$P < 0.01$; ****$P < 0.0001$. **(C)** Tubulin intensity fraction normalized to nontransfected cells (NT) in cells expressing Spastin-RFP with GFP, Rtn-1A-GFP, or Rtn-1C-GFP compared with nontransfected cells. Data are displayed as the mean ± SE from 145 nontransfected cells, 33 Spastin-RFP/GFP cells, 35 Spastin-RFP/Rtn-1A-GFP cells, and 32 Spastin-RFP/Rtn-1C-GFP cells from N = 2 biological replicates. One-way ANOVA with Tukey's post hoc test; *$P < 0.05$; **$P < 0.01$; ***$P < 0.001$; ****$P < 0.0001$. **(D)** Domain organization of Spastin isoforms (M1 and M87) and Rtn-1 isoforms (Rtn-1A and Rtn-1C), including the Reticulon homology domain (RHD), Microtubule Interacting and Trafficking domain (MIT), microtubule-binding domain (MTBD), and ATPases Associated with diverse cellular Activities (AAA) domain (severing domain). **(E)** Predicted binding affinities (ΔG binding, kcal/mol) between Spastin M87 and Rtn-1 isoforms at different domains, highlighting strong interactions with Rtn-1C. **(F)** Structural model (VMD snapshots) of the predicted interaction between the MTBD (yellow) of Spastin M87 (blue) and the N-terminal region (green) of Rtn-1C (red).

used a Rtn-1 siRNA that is directed against the conserved RHD sequence; thus, it also targets the Rtn-1A isoform. For instance, we observed limited co-distribution of Rtn-1A with Spastin compared with Rtn-1C, and its N terminus recruits other signaling pathways (Kaya et al, 2013). Therefore, the observed differential response might be due to the predominant presence of Rtn-1C in axons, whereas both isoforms are present in the cell bodies. Secondly, increased microtubule severing after axonal Rtn-1 KD might promote outgrowth when coupled to predominantly outward-directed growing filaments that characterize axons, but not to the radial or mixed orientation of microtubules in the cell bodies and dendrites (Baas et al, 2016). Interestingly, global KD of Rtn-1 in developing cortical neurons decreases axonal outgrowth (Nozumi et al, 2009), further supporting the relevance of understanding this compartmentalized response.

### The role of Spastin's microtubule severing on outgrowth after injury

The positive relationship that we observed between increased Spastin-severing activity and the outgrowth of injured axons may seem counterintuitive. However, microtubule severases, such as Spastin or Katanin, are known to increase the microtubule filament mass and growth under certain conditions. Indeed, under certain conditions Spastin can stabilize newly produced microtubule tips, paradoxically yielding a net increase in microtubule mass (Kuo & Howard, 2021). Furthermore, Spastin tends to concentrate at the tip of growing microtubules to behave as a dual enzyme that cleaves but also stabilizes them after cutting (Kuo et al, 2019). This stabilization property may also partially explain why we observed that the axonal inhibition of Spastin by SPTZ alone tends to promote outgrowth after injury (Fig 5C and D).

Like other studies, we observed that increased axonal outgrowth correlates with dynamic microtubule parameters such as growth rate (Liz et al, 2014). However, others have shown that the dynamic properties of microtubules, including microtubule growth rate and track length, can adapt to different outgrowth scenarios, with low capacity to predict axonal extension (Kiss et al, 2018). Thus, controlling dynamic parameters of microtubules is not the only factor that determines the outgrowth capacity of injured axons. The exact mechanisms by which locally synthesized Rtn-1C controls dynamic parameters of microtubules remain unsolved as they seem not to rely on Spastin-severing activity. Other Spastin-independent interactions of Rtn-1 may depend on its presence on the tubular ER, which includes the control of functional contacts with mitochondria and autophagosomes (D' Eletto et al, 2019; Reali et al, 2015), but their implications for microtubule dynamic parameters and axonal extension also remain to be solved.

Decreasing axonal Rtn-1C translation or increasing Spastin microtubule severing locally in axons may constitute an emerging mechanism for promoting injury recovery. For instance, local application of recombinant Spastin together with polyethylene glycol increases recovery after sciatic nerve injury (Lin et al, 2019). Furthermore, local augmentation of axonal Spastin by the pharmacological local regulation of 14-3-3 promotes the recovery of spinal cord injury (Liu et al, 2024). Indeed, axonal regeneration is particularly sensitive to the absence of Spastin (Stone et al, 2012). Similarly, results in *Drosophila* show that Spastin gene dosage is particularly relevant for the regenerative response of axons (Stone et al, 2012).

### Implications for length-dependent axonopathies and injury response

ER-shaping proteins participate in the etiology of multiple length-dependent axonopathies such as hereditary spastic paraplegias (HSPs) (Blackstone et al, 2011). Here, we have described an emerging mechanism relating Rtn-1C with the activity of Spastin, which is the most frequently mutated isoform in HSP (Hazan et al, 1999; Mannan et al, 2006). As a second point of analysis, most of the studies performed to understand axonopathies and axonal damage have focused on a "cell body–centered view" for protein synthesis and ER function. However, increasing evidence supports the relevance of axon-autonomous response to injury or cues from the local brain microenvironment (Lopez-Verrilli et al, 2013; Zhang et al, 2013, 2015; Jung et al, 2014). Similarly, our work contributes to highlight the autonomous response of cortical axons with potential implications during degeneration and injury.

# Materials and Methods

### Animals and surgical procedures

Pregnant Sprague Dawley rats at embryonic day 18 were obtained from the Central Animal Facility of Universidad Católica de Chile and euthanized in accordance with the guidelines of the Agencia Nacional de Investigación y Desarrollo de Chile (ANID). All the experimental procedures were approved by the Institutional Bioethics Committee of Universidad de los Andes (CEC2022106).

### Neuron cell cultures

Cortical neurons were cultured from 18-d embryos like previously reported (Deitch & Banker, 1993). Dissociated primary cortical neurons (20,000–30,000 cells) were seeded on glass coverslips coated with 0.1 mg/ml poly-L-lysine (P4707; Sigma-Aldrich). Microfluidic chambers were prepared as previously reported (Taylor et al, 2005). Chambers harboring axonal and somatic compartment connected with 750-µm-long microgrooves were mounted over a squared 25-mm glass coverslip previously coated with 0.1 mg/ml poly-D-lysine (354210; Corning) and 2 µg/ml laminin (L2020; Sigma-Aldrich). Plating medium was completely replaced with growth medium 24 h after seeding. Plating medium contained MEM (15630080; Gibco), 10% horse serum (16050122; Gibco), 0.6% D-glucose (G8769; Sigma-Aldrich), 0.1% GlutaMAX (35050061; Gibco), and 2% penicillin/0.1% streptomycin (15140122; Life Technologies), whereas growth medium contained Neurobasal (21103049; Gibco), 3% B27 (17504044; Gibco), 0.1% GlutaMAX, and penicillin/streptomycin 0.1%. An amount of 100 µl of fresh growth medium was added with a frequency of 48 h to the somatic compartment to prevent loss of water. Neuronball cell culture was performed as previously reported (Sasaki et al, 2010). Neurons were suspended in growth

medium supplemented with 10% horse serum. A total amount of 30,000 cells was seeded as hanging drops of 20 $\mu$l during 3 d inside the cover of 100-mm dishes filled with water. Floating spheres were gently washed three times to discard nonattached cells and then seeded on poly-L-lysine–coated culture plates. Each well contained 400 $\mu$l of growth medium previously supplemented with 1 $\mu$M cytosine $\beta$-D-arabinofuranoside hydrochloride (AraC, C1768; Sigma-Aldrich) to prevent glial growth. Neurites were allowed to grow until 7 d after seeding. The periphery of each neuronball was carefully scraped with a pipette and washed three times with cold PBS to further remove any remaining cell bodies. Cell body–containing spheres were extracted by vacuum suction.

## Immunofluorescence

Cells were washed twice with PBS and fixed with 4% PFA (P6148; Sigma-Aldrich) and 4% sucrose in PBS for 20 min at RT. The same procedure was applied to COS-7 cells. For assessment of growth cone morphology, neurons were fixed for 20 min at RT by adding an equal volume of 8% PFA and 22% sucrose in PBS directly to the culture medium. After fixation, cells were washed twice with PBS and sequentially incubated for 10 min each at RT with: (i) permeabilization solution (0.5% NP-40 [ab142227; Abcam] and 0.5% wt/vol BSA [BM-0150; Winkler] in PBS); (ii) quenching solution (50 mM $NH_4Cl$ in PBS); and (iii) blocking solution (0.5% wt/vol BSA and 5% FBS [F2442; Sigma-Aldrich] in PBS). Primary antibodies, diluted in 0.5% wt/vol BSA in PBS, were incubated overnight at 4°C. Coverslips were then washed three times (5 min each) with the same solution and incubated with secondary antibodies, diluted in the same buffer, for 1 h at RT. Finally, samples were washed with PBS and distilled water and mounted using VECTASHIELD.

## Growth cone morphometry

Growth cones (GCs) were imaged after $\beta$3-tubulin and F-actin labeling with phalloidin using identical acquisition settings across conditions. Analyses were performed in FIJI (ImageJ). The actin outline was thresholded (same parameters for all images) to isolate single binarized GCs at distal axon tips (axon shaft excluded). From the binarized GC mask, we computed area ($\mu m^2$), perimeter ($\mu m$), and elongation ratio (major axis/minor axis of the best-fit ellipse to the actin boundary). For skeleton-based metrics, the binarized GC mask was skeletonized (FIJI Skeletonize) to obtain a one-pixel-wide graph of the actin scaffold. We then used Analyze Skeleton (2D/3D) to extract (i) LSP (longest shortest path), defined as the maximum geodesic distance (in $\mu m$) between any two skeleton nodes within the GC—that is, the longest of all pairwise shortest paths along the skeleton—and (ii) number of filopodia, defined as the count of terminal branches (endpoints in the skeleton) that extend from the GC body and exceed a minimum length of 1.0 $\mu m$ (same length threshold applied to all conditions). When needed, small spurious branches produced by noise were pruned before counting. All measurements were performed blinded to condition.

## Colocalization analysis

COS-7 cells were transfected with indicated plasmids for 24 h and imaged using a confocal microscope (Fluoview FV1000; Olympus) equipped with a UPLSAPO 60×/1.35 NA objective and 4× digital zoom. Images were acquired with a Kalman filter (two iterations). For each condition, an average of 8–12 confocal sections (0.2 $\mu m$ thickness each) were analyzed per cell. Colocalization between RFP and GFP signals was quantified using Manders' coefficients (M1 and M2) with the JaCoP (Just Another Colocalization Plugin) in ImageJ (FIJI). Identical thresholding parameters were applied to each channel across all conditions.

## Fluorescence intensity analysis of COS-7

COS-7 cells were imaged with a Leica TCS SP8 confocal microscope using a 40×/1.30 NA oil-immersion objective. For each field, z-stacks of 15–17 optical sections (1 $\mu m$ step size) were acquired, and summed intensity projections were generated for analysis. Acquisition parameters (laser power, detector gain, and offset) were kept constant across all conditions. Cells were manually segmented in FIJI using $\alpha$-tubulin immunostaining to delineate them. For each ROI, mean fluorescence intensity of $\beta$3-tubulin, GFP, and RFP channels was measured, and background signal (from a cell-free region of the same image) was subtracted. Corrected mean intensity was multiplied by the cell area to calculate total fluorescence per cell. Fold change values were obtained by normalizing to the mean of the control group.

## Fluorescence intensity analysis of somas and axons

Images were acquired using an Olympus BX61WI microscope in wide-field configuration equipped with a 40×/1.30 NA oil-immersion objective and a Hamamatsu ORCA-R2 CCD camera, controlled by Cell[R] software (Olympus). Exposure time, gain, binning, and illumination settings were kept constant across all conditions. For somatic quantifications, cell bodies were manually segmented in FIJI using $\beta$3-tubulin staining as a guide, restricting the region of interest (ROI) to the soma and excluding processes. For each cell, the mean background intensity (measured in a cell-free region of the same image) was subtracted from the mean somatic intensity, and total fluorescence (integrated intensity) was calculated by multiplying by the corresponding cell area. Fold change per cell was determined by normalizing to the mean value of the corresponding control group. For axonal quantifications, a 1-$\mu m$-wide segmented line was drawn starting at the axon tip and extending 50 $\mu m$ proximally to extract the intensity profile (FIJI). Background intensity was subtracted from each axon profile using values measured in axon-free areas of the same image. For quantifying protein changes, the integrated intensity across the entire 0- to 50-$\mu m$ segment was calculated and expressed as fold change relative to controls. To minimize intensity variability across different experimental dates, while preserving the relative differences between conditions, we plotted the corrected intensity profiles. For $\beta$3-tubulin, a per-axon baseline was defined by subtracting the intensity at 0 $\mu m$ (the axon

tip) from the entire profile. For Rtn-1, the baseline was defined as the mean pixel intensity within the distal 5 $\mu$m of the profile (45–50 $\mu$m from the tip), which was subtracted from the entire trace of each axon. These normalizations were applied solely for visualization purposes and were not used in fold change calculations.

## Stimulated emission depletion (STED) microscopy analysis of axons

Super-resolution imaging was performed on a TCS SP8-3X STED microscope system (Leica Microsystems) equipped with a HC APO CS2 100x/1.40 oil objective. Distal axons of microfluidic chambers were imaged using excitation and depletion laser lines appropriate for each fluorophore (see Specific reagents). Images were digitized in 1,024 × 1,024 pixels format file with 5x zoom factor. These settings achieved 18.93-nm resolution in the XY plane and a 220-nm resolution in Z. Four confocal slices from each axon were sum-projected and analyzed. No deconvolution or post-processing algorithms were applied. Instead, all quantitative and qualitative analyses were performed directly on the raw STED images. Acquisition parameters (laser power, detector gain, pixel size, and dwell time) were kept constant across conditions. For quantification, the total fluorescence intensity of the last 15 $\mu$m of each distal axon was measured after background subtraction. Two analyses were performed: (i) intensity fold change was calculated by comparing nonaxotomized and axotomized axons; and (ii) the relative intensity, which was obtained by dividing the intensity of each pixel by the total intensity of the corresponding staining.

## Transfections

All transfections were performed Lipofectamine 3000 (L3000001; Thermo Fisher Scientific) following the manufacturer's instructions but without using p3000 reagent. For neurons and COS-7 cells, RNA or DNA complexes were added to chambers diluted in pure Neurobasal medium. For axonal transfection of siRNAs, medium on the axonal compartment was carefully removed and replaced with 50 $\mu$l of the transfection mixture containing 20 pmol of the corresponding oligonucleotide plus 50 $\mu$l of fresh growth media. ~300 $\mu$l of conditioned medium remained at the somatic compartment, generating positive flux toward the axons. Conversely, for somatic transfection the medium present at the somatic compartment was carefully removed and replaced with 50 $\mu$l of the transfection mixture plus 100 $\mu$l of fresh growth media, this time generating positive flux toward the soma. For validating siRNA efficiency, 200,000 neurons were seeded on 35-mm plates and transfected with 40 pmol of corresponding oligos and submitted to Western blot after 48 h. For colocalization, Western blot, and immunofluorescence analyses, COS-7 cells were seeded at ~1–2 × 10$^5$ per 25-mm coverslip and cultured for 24 h before transfection with 500 ng of pCMV3-C-GFPSpark plasmid encoding either Rtn-1C-GFP or Rtn-1A-GFP, together with pCMV3-C-ORFSpark encoding Spastin-RFP (all plasmids are detailed in the Specific reagents section).

## Axotomy and Sholl analysis

At 8 DIV, axons were removed from the axonal compartment of microfluidic chambers by vacuum aspiration performed five times with 100 $\mu$l of pure Neurobasal medium. Subsequently, all media were removed, and the transfection mixture was added either to the axonal or to the somatic compartment. For quantification of axonal outgrowth, $\beta$3-tubulin–stained axons were identically thresholded across conditions, binarized, and curated to fully trace axonal trajectories. The Sholl analysis was performed in FIJI using the NeuroAnatomy plugin. Concentric semicircles were applied at 2-$\mu$m radial intervals, starting from the ends of the microgrooves and centered on axonal fields of equivalent size. The analysis extended to a maximum radial distance of either 1,200 $\mu$m or 1,500 $\mu$m, depending on the experiment.

## Probe labeling of primary antibodies for puromycylation proximity ligation assay (puro-PLA)

Monoclonal antibodies mouse anti-Reticulon-1C and mouse anti-puromycin (3RH1, Kerafast) were directly labeled with the Duolink In Situ Probemaker PLUS (1002748995; Sigma-Aldrich) and MINUS (1002748992; Sigma-Aldrich), respectively, following the manufacturer's instructions. Both antibodies with a concentration of 1 $\mu$g/$\mu$l were labeled overnight at RT with the corresponding buffer and probes to a final volume of 20 $\mu$l. Labeled antibodies were further diluted with 20 $\mu$l of storage solution.

## Puro-PLA staining protocol

Puro-PLA was under conditions similar to well-established protocols (Dieck et al, 2015). Briefly, a 15-min pulse of 1.8 $\mu$M puromycin (P8833; Sigma-Aldrich) was applied exclusively to the axonal or somatic compartment of microfluidic chambers containing 9 DIV cortical neurons. As an additional control, this incubation was performed after 90 min of local pretreatment with 40 nM emetine (E2375; Sigma-Aldrich) in distilled water (vehicle), added either to the somatic or to the axonal compartment to block protein synthesis. Immediately after incubations, cells were washed twice with PBS and fixed with 4% PFA and 4% sucrose in PBS for 20 min at RT, followed by three PBS washes (5 min each). Cells were then permeabilized with 0.25% Triton X-100 in PBS for 10 min, washed three times with PBS (5 min each), blocked with 5% goat serum in PBS for 1 h, and incubated overnight at 4°C with probe-labeled antibodies (anti–Reticulon-1C, 1:50; anti-puromycin, 1:200) diluted in the same blocking solution. Detection of locally synthesized Reticulon-1C was performed following the manufacturer's instructions, except that no labeled secondary antibody step was included. Afterward, cells were incubated for 1 h at RT with anti-$\beta$3-tubulin conjugated to Alexa Fluor 488 and mounted with Duolink In Situ Mounting Medium with DAPI (Sigma-Aldrich). Images were acquired as previously described using a Zeiss LSM 800 confocal microscope with a 63× oil objective. PLA-positive puncta present within a $\beta$3-tubulin mask that defined the axonal area were quantified using FIJI software.

## RT–PCR from neuronballs and microfluidic chambers

For axonal RNA isolation from neuronballs, at least 200 balls were desomatized and processed with RNAqueous-Micro Total RNA Isolation Kit (AM193; Thermo Fisher Scientific), following the manufacturer's instructions. Approximately 20–30 axotomized neuronballs (somatic RNA) were processed using the same protocol. Reverse transcription was performed from 100 ng of RNA as input using the High-Capacity cDNA Reverse Transcription Kit (4368814; Thermo Fisher Scientific). For negative controls, axonal samples were submitted to the same protocol, but without adding reverse transcriptase. Resulting cDNA products were treated with DNase according to the manufacturer's instructions (EN0521; Thermo Fisher Scientific). For PCR, cDNA was amplified using the GoTaq Green Master Mix, 2X (M712; Promega), with the following cycling parameters: initial denaturation at 95°C for 5 min; 35 cycles of denaturation at 95°C for 30 s, annealing at 57°C for 30 s, and extension at 72°C for 45 s; followed by a final extension at 72°C for 10 min. PCR products were loaded on a 2% agarose gel and stained with GelRed Nucleic Acid Gel Stain (41003; Biotium), following the manufacturer's recommendations. We performed the RNA isolation from the axonal/somatic compartments of microfluidic chambers (7–8 chambers) as previously reported (Taylor et al, 2005). For the subsequent steps, we followed the same procedures as in neuronballs, but performing 42 PCR cycles.

## Western blot

Proteins were isolated using RIPA buffer (50 mM Tris–HCl, pH 7.4, 150 mM NaCl, 0.25% sodium deoxycholate, 1% NP-40, 1% SDS, 1 mM EDTA) supplemented with protease and phosphatase inhibitors (A32963; Thermo Fisher Scientific). Extracted proteins were quantified using the bicinchoninic acid (BCA) method (23225; Thermo Fisher Scientific) and separated on polyacrylamide gels under denaturing conditions (SDS–PAGE). For each sample, 20 $\mu$g of protein was loaded per lane for both complete neuronal or cell homogenates, and axonal or somatic fractions of neuronballs. Proteins were resolved on 15% acrylamide gels by electrophoresis at 70 V for 45 min followed by 100 V for 45 min. Gels were transferred to nitrocellulose membranes (Bio-Rad) at a constant current of 350 mA for 90 min. After transfer, membranes were blocked with 5% (wt/vol) skim milk in PBS for 1 h at RT under constant agitation and then incubated overnight at 4°C with the corresponding primary antibody diluted in PBS. Membranes were incubated with the appropriate secondary antibody for 1 h at RT. When indicated, total protein loading was normalized using No-Stain total protein labeling reagent (A44449; Thermo Fisher Scientific) according to the manufacturer's instructions. Bands and lanes were quantified by densitometry using Photoshop 7.0 (Adobe).

## Live imaging of microtubule plus ends

Cortical neurons were cultured in microfluidic chambers and transfected with EB3-Emerald GFP plasmid at 7 DIV. At 8 DIV,

neurons were axotomized by vacuum aspiration, and siRNAs were transfected into the axonal compartment. The Spastin inhibitor Spastazoline (HY-111548; MedChemExpress) was added to the axonal compartment 4–5 h after axotomy at a final concentration of 50 nM. Forty-four hours after transfection, axons expressing EB3-GFP were randomly selected and imaged in Tyrode's solution (125 mM NaCl, 2 mM KCl, 2 mM CaCl$_2$, 2 mM MgCl$_2$, 30 mM glucose, 25 mM Hepes, pH 7.4) at 32°C. Cells were equilibrated for 10 min before recording. Movies were acquired for 3 min at two frames per second (fps). To maintain hydrostatic pressure toward the axonal side, the volume difference between compartments was carefully preserved throughout the experiment. Imaging was performed on an inverted microscope (Observer.D1, Zeiss) equipped with an EMCCD camera (Evolve 512, Photometrics) controlled by VisiView software (Visitron Systems GmbH). A 63× oil-immersion objective and a GFP single-band exciter ET filter set (excitation 470/40, emission 590/22 nm, dichroic 59022BS) were used. The mEmerald-EB3-C-20 plasmid was a gift from Michael Davidson (plasmid #54076; http://n2t.net/addgene:54076; RRID: Addgene_54076; Addgene).

## Analysis of microtubule dynamics

All acquired time series were filtered using the Gaussian Blur in FIJI (applied to temporal frames). Mobile components of the series were isolated by applying a four-sigma Gaussian blur, whereas static elements were extracted by applying a 50-sigma blur. To enhance detection of moving structures, static components were subtracted from mobile components using the *Image Calculator* function in FIJI. The resulting filtered image series were then used to generate kymographs with the FIJI plugin *Kymograph Builder*. Kymographs were constructed from the distal portion of axons by tracing a 1-$\mu$m-wide line, with a total length ranging from 30 to 50 $\mu$m. Growth cones were excluded from the analysis. All analyzed parameters were quantified as previously described (Stepanova et al, 2010). EB3-GFP comets were analyzed from kymographs in FIJI. Individual dashes corresponding to microtubule growth events were manually traced using the Segmented Line tool. The angle of each dash was measured relative to the x-axis. The horizontal projection of the dash, obtained by multiplying its length by the cosine of the angle, represents the distance traveled by the comet in micrometers (track length). The vertical projection, obtained by multiplying the dash length by the sine of the angle, was proportional to comet's lifetime. To obtain comet's lifetime in seconds, vertical projection was divided by the total y-axis length of the recording kymograph and multiplied by the total acquisition time (180 s for a 3-min movie). The growth rate was calculated by dividing the track length by the comet's lifetime for each dash.

## Molecular docking between Spastin (M87) and Rtn-1C

First, Spastin M87 and Rtn-1A models of *Rattus norvegicus* were retrieved from AlphaFold Protein Structure Database (accession numbers AF-A0A0G2K590 and AF-Q64548, respectively). Then, M87 Spastin models were generated from the removal of the

first 86 amino acids of the M1 Spastin model, whereas Rtn-1C 3D-structure was built with MODELLER version 10.4 using an Rtn-1A model as a template. For this, amino acid sequences of Rtn-1C were retrieved from UniProtKB/Swiss-Prot (accession numbers Q64548). In addition, the first 20 amino acids (of the N-terminal region) of the Rtn-1C model and region 200–317 containing the microtubule-binding domain of Spastin M87 were refined using ModLoop server (Fiser & Sali, 2003). Subsequently, we evaluated the interaction of the Rtn-1C N terminus with three domains of Spastin M87—endosome or MIT (region 116–194), microtubule-binding (region 270–328), and microtubule-severing domain of the AAA ATPase (region 342–599)—using the HADDOCK docking protein–protein web server (Van Zundert et al, 2016). Among many models, low-energy conformations for each complex were extracted from HADDOCK 2.2 server, as previously described (Acevedo et al, 2018). The binding energy (ΔG_bind) of the complexes was predicted using the PRODIGY web server at a temperature of 310 K, and chain identifiers for Rtn-1 and Spastin M87 were specified as Interactor 1 and 2, respectively. Finally, graphical analysis of molecular docking studies was performed using VMD 1.9.2 software (Humphrey et al, 1996).

**Specific reagents**

| Antibody | Cat. No | Company | Host | Description | Dil. IF | Dil. WB |
|---|---|---|---|---|---|---|
| Primary antibodies | | | | | | |
| Reticulon-1C | sc-23882 | Santa Cruz | Mouse | | 1:50 | 1:500 |
| Reticulon-1C | ab8961 | Abcam | Mouse | Proximity ligation assay and Western blot | 1:1,000 | — |
| Reticulon-1A | AB 9274 | Abcam | Mouse | | 1:500 | |
| Reticulon-1 (all isoforms) | ab83049 | Abcam | Rabbit | | 1:500 | 1:1,000 |
| Spastin | PA5-53581 | Invitrogen | Rabbit | | 1:500 | — |
| KDEL | PA1-013 | Thermo Fisher Scientific | Rabbit | | 1:200 | |
| MAP2 | M3696 | Sigma-Aldrich | Rabbit | | — | 1:500 |
| MAP2 | MAB3418 | Millipore | Mouse | | 1:500 | — |
| β3-tubulin | NB100-1612 | Novus | Chicken | | 1:2,000 | |
| β3-tubulin | 2G10 | Abcam | Mouse | Neuronball Western blots | — | 1:5,000 |
| NF-M | AB 1987 | Millipore | Rabbit | | 1:500 | |
| Puromycin | 3RH1 | Kerafast | Mouse | | 1:200 | — |
| Tau | MAB3420 | Chemicon | Mouse | | | 1:1,000 |
| Phalloidin | A12379 | Thermo Fisher Scientific | — | Growth cone staining | 1:500 | |
| Secondary antibodies | | | | | | |
| Alexa Fluor 488 anti-mouse | a21202 | Thermo Fisher Scientific | Donkey | | 1:1,000 | — |
| Alexa Fluor 555 anti-rabbit | a21429 | Thermo Fisher Scientific | Goat | | 1:1,000 | — |
| Alexa Fluor 488 anti-rabbit | a11034 | Thermo Fisher Scientific | Goat | | 1:1,000 | — |
| Alexa Fluor 647 anti-chicken | A-21449 | Thermo Fisher Scientific | Goat | | 1:1,000 | — |
| Anti-rabbit IgG—Atto 594 | 77671 | Sigma-Aldrich | Goat | Used for experiments with STED | 1:500 | — |
| Anti-mouse—Atto 488 | 62197 | Sigma-Aldrich | Goat | Used for experiments with STED | 1:500 | — |
| Anti-mouse anti-IgG | 31430 | Thermo Fisher Scientific | Goat | HRP-conjugated antibody | — | 1:5,000 |
| Anti-rabbit IgG | 31460 | Thermo Fisher Scientific | Goat | HRP-conjugated antibody | — | 1:5,000 |
| Anti-chicken IgG | SAB3700195 | Sigma-Aldrich | Goat | HRP-conjugated antibody | — | 1:5,000 |

**Plasmids**

| Gene | Backbone | Reporter | Promoter |
|---|---|---|---|
| Rat RTN1A Gene ORF (NM_053865.1) | pCMV3-C-GFPSpark | GFP | CMV (overexpression) |
| Rat RTN1C Gene ORF (XM_008764690) | pCMV3-C-GFPSpark | GFP | CMV (overexpression) |
| Rat SPAST Gene ORF (NM_001108702.2) | pCMV3-C-ORFSpark | ORF | CMV (overexpression) |
| pCMV3-C-GFPSpark Control Vector | pCMV3-C-GFPSpark | GFP | CMV (overexpression) |
| Rat EB3 Gene ORF (NM_012326.2) | mEmerald-C1 | mEmerald-C1 (GFP) | CMV (overexpression) |

**Primers**

| Gene | Forward | Reverse |
|---|---|---|
| H1 histone | 5'-ACCCATTGTTCAAGGACAGC-3' | 5'-ATCAGGTCCCCCAACTTACC-3' |
| β-actin | 5'-AGCCATGTACGTAGCCATCC-3' | 5'- CTCTCAGCTGTGGTGGTGAA-3' |
| Reticulon-1 | 5'-GAGCAGATCCAGAAGTACAC-3' | 5'-GAAACCACAGCCATAAGCAG-3' |
| Reticulon-1A | 5'- ATGGAAACTGCATCCACA-3' | 5'-AAAAGTATGCAGAGTCCTC-3' |
| Reticulon-1C | 5'-ACCATGCAGGCCACTGCCGATTC-3' | 5'-CTCAGCGTGCCTCTTGGCGCC-3' |
| GFAP | 5'-GGCGCTCAATGCTGGCTTCA-3' | 5'-TCTGCCTCCAGCCTCAGGTT-3' |

**Oligonucleotides**

| Oligo | +Strand | −Strand |
|---|---|---|
| siRNA 1 | 5'rGrArArArArGrUrCrArGrGrCrUrArUrUrGrArCrCrUrUrCrTrG3' | 5'rCrArGrArArGrGrUrCrArArUrArGrCrCrUrGrArCrUrUrUrUrCrCrA3' |
| siRNA 2 | 5'rCrGrArUrGrUrUrUrArCrUrCrUrrArCrCrUrGrGrGrGrUrATA3' | 5'rUrArUrArCrCrArCrArGrGrUrArArrArCrArUrCrGrArA3' |
| siRNA 3 | 5'rGrUrGrGrUrUrUrCrGrArUrGrUrUrUrArCrUrCrUrArCrCTG3' | 5'rCrArGrGrUrArGrArGrUrArArArCrrArUrCrGrArArArCrCrArCrArG3' |
| IDT Negative Control DsiRNA (51-01-14-03) | | |

## Statistical analysis

Unless otherwise stated, data are presented as the mean ± SE. Normality was assessed using the Shapiro–Wilk test. For normally distributed data, comparisons between two groups were performed using a two-tailed *t* test; for non-normally distributed data, the two-tailed Mann–Whitney *U* test was applied. Western blot fold change values were compared with a theoretical mean of 1 using the Wilcoxon signed-rank test. For comparisons among multiple groups, one-way ANOVA followed by Tukey's post hoc test was used for normally distributed data, whereas the Kruskal–Wallis test followed by Dunn's post hoc test was applied for nonparametric data. Statistical significance was defined as $P < 0.05$. Fluorescence intensity experiments in Figs 2 and 4 were analyzed at the single-cell level using linear mixed-effects (LME) models, with experimental date included as a random effect. Statistical tests were performed using GraphPad Prism (GraphPad Software) and R (v4.0 or later) with the lme4 and lmerTest packages.

## Data and Materials Availability

The datasets generated and/or analyzed in this study (raw and processed imaging/biochemical measurements from cell-based experiments, and associated metadata) are available from A Luarte (aluarte@uandes.cl) upon reasonable request. Public deposition is not currently feasible because the complete dataset is distributed across large raw files and requires accompanying experimental metadata/organization for correct reuse.

## Supplementary Information

## Acknowledgements

This research was partially supported by HPC OCÉANO (FONDEQUIP No EQM170214) and the supercomputing infrastructure of the NLHPC (CCSS210001), ANID Basal funding for Scientific and Technological Center of Excellence, IMPACT, FB210024, and Centro Nacional de Inteligencia Artificial (CENIA), FB210017.

### Author Contributions

A Luarte: conceptualization, resources, supervision, investigation, visualization, and writing—original draft, review, and editing.
J Gallardo: formal analysis and investigation.
D Corvalán: investigation and methodology.
A Chakraborty: investigation and methodology.
C Gouveia-Roque: investigation and writing—review and editing.
F Bertin: investigation and methodology.
C Contreras: investigation and methodology.
JP Ramírez: data curation and visualization.
A Weber: methodology.
W Acevedo: data curation and software.
W Zuschratter: resources, supervision, investigation, visualization, and methodology.
R Herrera-Molina: investigation and writing—review and editing.

U Wyneken: funding acquisition and writing—review and editing.

A Paula-Lima: conceptualization, supervision, funding acquisition, methodology, and writing—review and editing.

T Adasme-Rocha: investigation.

J Toledo: data curation, investigation, and methodology.

R Vergara: data curation, software, and formal analysis.

A Figueroa: investigation and methodology.

C González: funding acquisition, investigation, visualization, and writing—review and editing.

C González-Billault: investigation, visualization, and writing—review and editing.

U Hengst: resources, investigation, methodology, and writing—original draft, review, and editing.

A Couve: resources, funding acquisition, investigation, visualization, methodology, and writing—original draft, review, and editing.

## Conflict of Interest Statement

The authors declare that they have no conflict of interest.

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
