## [Reviewer comments · Life Science Alliance]

Reticulon-1 synthesis controls outgrowth and microtubule dynamics in injured cortical axons

Alejandro Luarte, Javiera Gallardo, Daniela Corvalan, Ankush Chakraborty, Claudio Gouveia-Roque, Francisca Bertin, Carlos Contreras, Juan Ramirez, Andre Weber, Waldo Acevedo, Werner Zuschratter, Rodrigo Herrera Molina, Ursula Wyneken, Andrea Paula Lima, Tatiana Adasme, Jorge Toledo, Rodrigo Vergara, Antonia Figueroa, Carolina Gonzalez, Christian González-Billault, Ulrich Hengst, and Andres Couve

DOI: <https://doi.org/10.26508/lsa.202503571>

Corresponding author(s): Alejandro Luarte, Universidad de Los Andes, Chile

Review Timeline:

Submission Date:	2025-11-17
Editorial Decision:	2025-11-24
Revision Received:	2025-12-08
Editorial Decision:	2025-12-17
Revision Received:	2025-12-29
Accepted:	2026-01-05

Scientific Editor: Tim Fessenden

Transaction Report:

Please note that the manuscript was previously reviewed at another journal and the reports were taken into account in the decision-making process at *Life Science Alliance*.

Review #1

1. Evidence, reproducibility and clarity:

In this paper, the authors focus on the role of Reticulon-1C in concert with Spastin in response to axonal injury. In data mining, they find axonal mRNAs encoding for ER-associated proteins including Rtn-1. They establish a knockdown targeting both Rtn-1 isoforms Rtn-1A and Rtn-1C. They observe decreased beta-3-Tubulin levels in the soma while axonal protein levels are unchanged. In microfluidic devices, they characterise the effect of a compartment-specific Rtn-1 KD on axonal outgrowth in the axonal compartment. The authors quantify axonal outgrowth, seeing increased outgrowth in an axonal compartment-specific Rtn-1 KD, while the effect seems to be reversed when applying the KD construct in the somatic compartment. When focussing on the axonal growth cone, they find the Rtn-1 KD shows differences in several morphological features of the growth cone. They find an increase in Tubulin levels in an axonal compartment-specific, but a decrease in a somatic compartment-specific Rtn-1 KD. Colocalisation of Rtn-1C and Spastin is shown to be monolaterally increased following axotomy. Combining axotomy with the Rtn-1 KD shows increases in dynamic microtubule growth rates and track lengths. In another model system, neuron balls, they show Rtn1-C, but not Rtn1-A to be present in the axon. In a puro-PL assay they also show it can be synthesised in the axonal compartment. To investigate the mechanism enabling the cooperation between Spastin and Rtn-1C, they move to a cell line model in which they see a correlating distribution between Spastin and Rtn-1C but not Rtn-1A. Finally, they use in silico modelling to speculate on binding between Spastin domains and Rtn-1 isoforms.

****Major comment:****

The rationale behind the work is convincing, however some interpretations are presented as more robust than some data allow. Most notably, while the interaction between Rtn-1 and Spastin has been shown prior to this study, it is only presented here through in silico analysis. In figure 5, an increase in the growth rate of dynamic microtubules is observed in either a Rtn-1C KD or by using a Spastin-inhibitor. Due to a described increase in colocalisation between Rtn-1C and Spastin (5A), the increase in growth rate is displayed as caused by Rtn-1 promoting Spastin's severing ability. This result might however be correlative. Further in the injured samples, Spastin-levels seemingly increase (in the representative images) and it is thus not surprising that the level of Rtn-1C colocalising with Spastin increases as well. This might not be indicative of a cooperation and further experimental evidence are required.

****Other comments:****

- Generally, graphs would benefit from individual values plotted as well as the summary. Font sizes and types (but rarely) are sometimes inconsistent. Proteins should be consistently written (capitalised or not).
- Table 1 and figure 1 present data collected from a vast amount of resources. It should be highlighted that datasets from which data was obtained includes many different models, different DIVs and neuronal cell types. Figure 1B may benefit from a different colour scheme. "Ex-vivo" should be "Ex vivo". For "ER mRNAs are a relevant category" it is not described what "relevant" would mean in this context. The title might remove this small part or describe it in the text. It should be described how it is decided that mRNAs are "common".
- Figure 2: add description to y-axis to describe what fold change is displayed, applies to multiple figures. Will improve readability of the figures. In 2C, the ROI showing neuronal somata should be increased to show part of the axon and not cut off the soma.
- Figure 3: Three out of four axonal compartments seem to be comprised of dying or damaged axons. Especially the axonal KD scrambled image. It should be ensured that neuronal cultures are healthy. Typo in "intersections". The schematic of 3B is a great addition to explain the graphs above. Perhaps it could be a bit refined as it is currently hard to see whether this is a neuron or a growth cone without context. Maybe show where the axon connects to the depicted growth cones and change the third icon which looks like it was crossed out. Small formatting issues: remove additional space bar before "Figure 3." And add after "Bar"
- Figure 4: If not misunderstanding what is depicted, in 4A and B, different lookup tables are used to depict the same signal. Only one of each images is necessary. Do the axons have more tiny branches in the Rtn-1 KD condition in 4A? Unclear why Rtn-1 levels are increased in the Rtn-1 KD (4C), please clarify.

- Figure 5: It may be easier to understand what "axotomy" samples are if just referred to as "injured" as later in the same figure. The procedure could also very briefly be explained in the results. 5C should depict AUC in μm^2 not μm . 5D Spastin is barely visible, brightness and contrast should be adjusted to enhance visibility.
- Figure 6: It should be made clear why it is necessary to switch to another model system just for 6A, please indicate this in the text. PCR bands seem very pixelated, check the quality. It is unclear why some genes/proteins were only tested with either PCR or WB others with both. Rtn-1C and Rtn1-A should be presented in the same order in the PCR and WB panel. Correct "Rtn1-1A" typo. In 6D, 1.5 dots per soma seems like a low number. When normalised to the area the soma vs the axon occupies, the compartmentalisation does not work? May be it make sense to refine analysis or apply puromycin in the somatic compartment and analyse the axonal compartment as comparison?
- Figure 7: 7A shows two images depicting the same information that may not be needed. Can probably be removed. In 7B there is no negative (or any) correlation between Spastin levels and Tubulin, however later it is mentioned that Rtn-1C transports Spastin thus causing a decrease in Tubulin at certain locations? It is unclear if Spastin levels vary intensely between different samples. Mean intensity of the somatic area may be beneficial to rule this out. 7B Tubulin on the right top panel seems to have a decrease in Tubulin levels which is not visible due to the Y axis of Tubulin being set to a different range than the middle and lower panel. The average of line scans from multiple cells may be helpful to determine whether there is indeed no colocalization between Rtn-1A and Spastin. The provided representative images seem to show similar degrees of colocalization between Spastin and Rtn-1A/C.

****Results:****

- It would be helpful to reiterate the hypothesis at the start to ease the reading flow.
- There seems to be minor redundancy in lines 132-138.
- There are several spellings, proof-reading is recommended. For example, in line 136 should be "promotes". 160 "locala", 192 should be "the actin cytoskeleton", 194 should be "we first examined", 195 should be "Different", 223 "using", 259 "axons". ...
- 154-155: Unclear, why the lower MW Rtn-1C was seen as more important.
- 167 results of 2E not stated before interpreting them.
- 181 would suggest "outline" instead of "perimeter".
- 183-184 "longest shortest path" is a confusing term.
- figure 4B should be referenced earlier in the sentence.
- 243-244 may be correlation. Rtn-1 and Spastin do not necessarily interact so that this result is achieved.
- 246: In figure 1 the KD seemed to have an effect on both Rtn-1 isoforms, why not here anymore? 259 "axons". 284 "counteract" instead of "suppress"?
- 485: rephrase as the interaction between Rtn-1C with Spastin has not been shown directly in these experiments.

Methods: 535 "in PBS". 543 citation error. 689-699 is it necessary to add a gaussian blur?

References: Mannan, A U et al. appears twice in the citation list (36 and 44).

2. Significance:

Overall, this manuscript describes novel findings which will be interesting to the neuronal cell biology community and scientists working on the field of neuronal injury and regeneration. It is well structured, and the data are mostly well presented but sometimes conclusions are over-interpreted. However, several points need to be addressed in a more convincing way.

Review #2

1. Evidence, reproducibility and clarity:

Axonal mRNA localization and localized translation supports many neuronal functions and is an important determinant of the regenerative potential of axons after injury. How this works mechanistically remains unclear. The authors present a well performed and technically challenging study in which they identify RTN-1 as a regulator of axonal outgrowth after injury. They provide evidence using experiments in microfluidic chambers that RTN1 is locally synthesized in axons. Interestingly, they identify a (local) interplay between RTN1 and Spastin which affects microtubules and thereby regulates the outgrowth of cortical axons after injury. This study provides an interesting new link between a locally synthesized protein (RTN1) and a microtubule-regulating protein Spastin that is changed upon axon injury. This provides an advance in our understanding in axon regeneration after injury and provides the basis for new studies that can further investigate this interplay. Although interesting, I have several concerns that should be

clarified and are needed to substantiate the findings and model presented in this study.

****Major concerns:****

1. In figure 1, the authors provide an analysis of overlapping axonal mRNAs. There are more axonal transcriptome studies and a recent study by von Kugelgen and Chekulaeva (2020; doi: 10.1002/wrna.1590) already performed such an analysis, which included more studies. It would be good to mention this. It can be perceived that studies were now chosen to get the outcome that Rtn-1 is present in all studies. For example, von Kugelgen finds mRNA coding for RTN3, another ER structural protein, as present in 16 out of 20 studies analyzed. That said, the authors present more reasons to look at Rtn-1, so the selection to continue with this protein remains valid but can be written up differently so not to present it as the 'sole' ER-shaping protein consistently present in axonal transcriptomes.
2. The description of methods is currently insufficient and incomplete and does not allow for reproducibility of this study. For example, different Rtn-1 antibodies seem to be used in this study. Is the same antibody used for staining and WB? There is no listing of any of the antibodies used in the study and which one is used for which technique/experiment. This should be clarified and should be easy to do so in the methods section (antibody name, origin/company, dilution used) to enhance reproducibility of this study. This is not limited to primary antibodies and any information on secondary antibodies, including what was used for STED is completely missing.
3. The timeline of KD experiments in Figure 2 and 3 are unclear. For the Western blot KD is performed at DIV7 and collected 48 hours later. However, this is not specified for the stainings done in Figure 2C-E. Is this also at DIV7 and then for 48 hours? In figure 3 the siRNA is added at DIV8 (together with axotomy) and outgrowth is measured 24 hours later. Is 24 hours sufficient to achieve knockdown? Is this also what was done for stainings? Later on in Figure 5B, 48 hours of KD is again used. It is unclear what the rationale of these differing timepoints is. Why was this chosen? Is the timeline also the reason for the difference in segment lengths chosen? In Figure 3, there is a significant effect on outgrowth in the KD in the 'mid-range' which is not present in Figure 5.
4. Could the authors provide a rescue condition for their siRNA (using a siRNA-resistant construct) to show that their siRNA is specific for RTN1. They nicely show the efficiency of the siRNA but not its specificity. This is crucial because if not specific, this will affect a large part of their study. They already have RTN1A and RTN1C constructs available. Such a rescue experiment should ideally also be performed for one or more of their phenotypic experiments, such as the one presented in Figure 3A or 5 to show that the phenotype is really RTN1 dependent. If done by re-expressing either RTN1A or RTN1C, this could provide insightful information on the relevant isoforms.
5. I find the data presented in Figure 4A/B confusing. Axonal RTN-1 KD does not reduce axonal RTN1 levels but somatic KD does. I understand that this implies most protein comes from the soma and the authors indeed present an explanation that increased somatic RTN1 occurs after axonal KD as a compensation mechanism. However, this can also be interpreted that there is no axonal synthesis of RTN1 after injury and axonal KD has indirect or even aspecific effects. Their model depends on this difference. Their data in Figure 6 could provide supporting evidence if it shows RTN1 puro-PLA after injury. Along these same lines, in Figure 6, they nicely include a compartment control for puro-PLA. It therefore seems doable to include a somatic puromycin control for their axonal puro-PLA, to exclude and diffusion/transport of the newly synthesized peptides. This is especially in light of two recent papers reporting on this possible phenomenon, although these studies were not performed in neurons.
6. In Figure 5A the authors find an increased co-localization (RTN1/Spastin) after axotomy. From their images, it seems that the amount of Spastin is hugely increased, which would by default increase the chance of (random) colocalization of RTN1 on Spastin. Could the authors comment on this?
7. In figure 5E and 5F, the condition of scr + SPTZ is omitted. What is the reason for this? The explanation of results in these figures is confusing. The authors report a 'clear trend' in increase in comet track length and lifetime upon addition of SPTZ to axonal RTN-1 KD. This is however not significant. The comparisons that are made afterwards are confusing (e.g. increase in comet lifetime of SPTZ in non-injured axons with RTN1 KD compared to Scr+DMSO and KD + DMSO in injured axons). Their conclusion is axonal RTN-1 synthesis in injured axons (see my concern in the points above on this) governs microtubules growth rate beyond Spastin activity yet blocking Spastin activity still completely blocks the effect of KD on outgrowth.

****Other/minor concerns:****

- The gene ontology analysis in Figure 1A contains the category 'Endoplasmic reticulum'. In this category are mainly ribosomal proteins. Although in a gene ontology analysis these proteins will be included in this category, it is misleading in this respect since they are just as likely to be coming from cytoplasmic ribosomes. Although it cannot be excluded that these are ER-bound ribosomes, not in the last place because a recent study (Koppers et al., 2024, doi:

10.1016/j.devcel.2024.05.005) found ribosomes attached to the ER in axons, I believe the category should be adapted or at the least clarified in the text.

- Is RTN-1C isoform still an ER-shaping protein or rather an ER protein with alternative functions? The final sentence in the abstract makes a statement that a locally synthesized ER-shaping protein lessens microtubule dynamics. Could the authors provide a clearer description and discussion of the evidence in literature for this? RTN1C has been suggested to perform alternative functions in which case the statement that the local synthesis of an ER-shaping protein is important for axonal outgrowth should be adapted.

- Is there a difference in RTN1 distribution or levels pre- and post-axotomy?

- Line 100/101 states 'the interactome of the axonal ER provides...'. To my knowledge there has been no study looking at the interactome of the axonal ER specifically. Surely axonal ER proteins are known but there is a difference.

- Typo line 160 'locala'

- In Figure S1 B, please add the DIVs to make it more clear what each graph corresponds to. The legend of S1B states different distances from the cell body but the graph shows distances from the tip.

- Figure 2C, why does B3 tubulin decrease in soma, aspecific effect of siRNA?

- What is the rationale on the opposite effect found in outgrowth in Figure 3?

- Missing word 'we' on line 194

- Typo line 629 'witmn h', please proofread the entire manuscript carefully.

- Could the authors comment on why, in Figure 7B/C, GFP only is colocalizing with Spastin-RFP? In general, GFP should be diffusive and not display punctate colocalization with Spastin.

2. Significance:

Axonal mRNA localization and localized translation supports many neuronal functions and is an important determinant of the regenerative potential of axons after injury. How this works mechanistically remains unclear. The authors present a well performed and technically challenging study in which they identify RTN-1 as a regulator of axonal outgrowth after injury. They provide evidence using experiments in microfluidic chambers that RTN1 is locally synthesized in axons. Interestingly, they identify a (local) interplay between RTN1 and Spastin which affects microtubules and thereby regulates the outgrowth of cortical axons after injury. This study provides an interesting new link between a locally synthesized protein (RTN1) and a microtubule-regulating protein Spastin that is changed upon axon injury. This provides an advance in our understanding in axon regeneration after injury and provides the basis for new studies that can further investigate this interplay. Although interesting, I have several concerns that should be clarified and are needed to substantiate the findings and model presented in this study.

The audience for this study will be mainly basic research in the fields of both axonal protein synthesis and axon regeneration. My expertise is in the field of mRNA localization and local protein synthesis.

Review #3

1. Evidence, reproducibility and clarity:

This manuscript investigates the relationship between the endoplasmic reticulum morphogen reticulon-1 (Rtn-1) and the microtubule severing protein spastin in axons after injury. The main message and conclusion of the paper is that local axonal synthesis of Rtn-1 plays a role in regulating the microtubule severing activity of spastin by interacting with spastin and inhibiting its activity. This mechanism would be important after injury by regulating axonal growth.

The conclusions of the paper are based on the following claims:

1. Rtn-1 is synthesized locally in axons.
2. Specific downregulation in Rtn-1 in axons using microfluidic chambers affects microtubules abundance (measured by beta-3 tubulin) and promotes axon growth after injury.
3. Inhibition of spastin MT-severing activity with a specific drug rescues the growth effect induced by axonal downregulation of Rtn-1.
4. Rtn-1c interacts with spastin-M87 to limit its MT-severing activity in a cellular system upon overexpression.

Major comments:

1. Evidence that Rtn-1 is synthesized in axons comes from two experiments. Initially, the authors show that Rtn-1 siRNA transfection in the axonal compartment of microfluidic chambers reduces Rtn-1 levels in axons, suggesting that there is some local synthesis. Although this method is very attractive, I am concerned about the statistical analysis.

The graphs show bars rather than individual data points from the average of a large number of neurons (about 300). The plots also show the SEM instead of the SD, thus covering all the variability that is inherent in this type of experiment. The statistics are probably not performed on the 3 biological replicates, but consider the individual neurons as N. This is obviously not correct, since neurons in an experiment may all be affected by the same technical problem and are not independent replicates. For this reason, I am a bit skeptical about this quantification. Another problem is that the quantification of the fluorescence intensity of the sample does not take the nuclei into account. Are the nuclei removed for analysis? Are the images single planes? Addressing the quantification issues is crucial also for data in Figure 4, where the authors show a different effect of Rtn-1 axonal KD after injury.

The second experiment is the Puro-PLA in Figure 6D. This experiment shows an average of 1.5 dots of signal per soma, which is a very low level of translation for this compartment where most of the synthesis should be taking place. In the axons, it is not clear how they calculate the axonal area. Again, the number of dots detected is very low and the physiological significance is questionable. A control with a known mRNA translated in axons would be important. Finally, as an important control, the authors should show the presence of Rtn-1 mRNA by FISH in their experimental system.

2. The effects on tubulin following Rtn-1 downregulation in axons is potentially very interesting, but the authors should be careful because it could also mean that the axons are suffering. Can they also stain for other cytoskeletal markers?

3. The results using SPTZ are very interesting and implicate spastin microtubule severing activity in the observed phenotype. In my opinion these experiments however do not prove that "axonal Rtn-1 is indeed promoting the severing of microtubules by spastin", but simply that the blocking spastin activity prevents the appearance of the microtubular phenotype (which appears still with a mysterious mechanism). What happens if they try to stabilize the cytoskeleton by another mean (with taxol for example?). The authors should rephrase this conclusion.

4. The last experiment (Figure 7) that aims to connect Rtn-1 and spastin function is very artificial, since it is based on overexpression. Why should spastin M87 interact with an ER morphogen? Endogenously it is conceivable that spastin M1 which localizes to the ER would interact with Rtn-1. Moreover, this experiment needs further controls and quantifications. First, it is quite obvious from panel 7C that there is crossover of signal in the two fluorescence channels (see GFP and spastin). Controls need to be shown, where only one of the two fluorescent proteins is expressed and the specificity of the laser is tested. This experiment is based on only 1 cell shown where co-localisation is detected based on a line that is placed in a specific area of the cell. The effects on the microtubular network needs quantification.

5. What is exactly the model proposed? The title implies that axonal synthesis of Rtn-1 is important during injury, but the data in the paper rather suggest that upon injury the majority of Rtn-1 is not locally synthesized. If the levels of Rtn-1 do not change, why the effect on the microtubules should be specific? Why would a siRNA against Rtn-1 in axons not affect the levels of Rtn-1, but those of tubulin? The authors should be careful, and test other control siRNAs, and Rtn-1 siRNAs, since it is well known even in more simple cellular systems that the toxicity of individual siRNAs can vary greatly.

****Minor comments:****

In Figure 5A, it would be helpful to indicate the border of the axon. The figure is not really convincing.

2. Significance:

The manuscript uses complex methods to address an interesting cell biological question of relevance to understand axonal growth regulation upon injury. A limitation of the study is the statistical analysis, which triggers some doubts about the reproducibility of the data. Further experiments and the addition of controls would be important to support the claims of the authors.

Response to reviewers

We thank the reviewers for their constructive feedback, which has greatly improved the clarity and rigor of our manuscript. We have carefully addressed each comment below, indicating changes made to the text, figures, or supplementary material where appropriate. References to line numbers correspond to the revised version of the manuscript.

Reviewer #1 (Evidence, reproducibility and clarity (Required)):

In this paper, the authors focus on the role of Reticulon-1C in concert with Spastin in response to axonal injury. In data mining, they find axonal mRNAs encoding for ER-associated proteins including Rtn-1. They establish a knockdown targeting both Rtn-1 isoforms Rtn-1A and Rtn-1C. They observe decreased beta-3-Tubulin levels in the soma while axonal protein levels are unchanged. In microfluidic devices, they characterise the effect of a compartment-specific Rtn-1 KD on axonal outgrowth in the axonal compartment. The authors quantify axonal outgrowth, seeing increased outgrowth in an axonal compartment-specific Rtn-1 KD, while the effect seems to be reversed when applying the KD construct in the somatic compartment. When focussing on the axonal growth cone, they find the Rtn-1 KD shows differences in several morphological features of the growth cone. They find an increase in Tubulin levels in an axonal compartment-specific, but a decrease in a somatic compartment-specific Rtn-1 KD. Colocalisation of Rtn-1C and Spastin is shown to be monolaterally increased following axotomy. Combining axotomy with the Rtn-1 KD shows increases in dynamic microtubule growth rates and track lengths. In another model system, neuron balls, they show Rtn-1C, but not Rtn-1A to be present in the axon. In a puro-PLA assay they also show it can be synthesised in the axonal compartment. To investigate the mechanism enabling the cooperation between Spastin and Rtn-1C, they move to a cell line model in which they see a correlating distribution between Spastin and Rtn-1C but not Rtn-1A. Finally, they use in silico modelling to speculate on binding between Spastin domains and Rtn-1 isoforms.

Major comment:

The rationale behind the work is convincing, however some interpretations are presented as more robust than some data allow. Most notably, while the interaction between Rtn-1 and Spastin has been shown prior to this study, it is only presented here through in silico analysis. In figure 5, an increase in the growth rate of dynamic microtubules is observed in either a Rtn-1C KD or by using a Spastin-inhibitor. Due to a described increase in colocalisation between Rtn-1C and Spastin (5A), the increase in growth rate is displayed as caused by Rtn-1 promoting Spastin's severing ability. This result might however be correlative. Further in the injured samples, Spastin-levels seemingly increase (in the representative images) and it is thus not surprising that the level of Rtn-1C colocalising with Spastin increases as well. This might not be indicative of a cooperation and further experimental evidence are required.

R: We thank the reviewer for this thoughtful comment. We agree that our interpretation should be more cautious, and we have revised the Title, Results and Discussion sections accordingly. In particular:

1. Following yours and other reviewer comments, we have analyzed a new set of experiments regarding the STED images of non-injured and injured axons. To eliminate the risk of artifactual descriptions, we have avoided deconvolution and worked directly with raw STED images (Figure 5A). Under these conditions, the distribution of Spastin and its intensity in distal axons are not modified by injury, nor those of Rtn-1C and Spastin (Supplementary figure 4). We emphasize in the revised text that the *in silico* modeling we present is supportive, but not definitive, of a direct interaction. To address this concern, we clarify that our study builds on prior evidence of biochemical interaction between Rtn-1C and Spastin (Mannan et al., 2006), and that our own data demonstrate: i) compatible subcellular distribution in axons by super-resolution (STED microscopy, Figure 5A); ii) a potential functional interplay in axons (rescue of β 3-tubulin levels by Spastin inhibition, Figure 5B), and iii) isoform-specific co-distribution with Spastin in heterologous cells that is associated with changes on microtubule integrity (see improved Figure 7). Together, these results go beyond correlative localization, but we acknowledge that they do not directly demonstrate a molecular complex in axons. Thus, we now indicate that “Although we did not directly test their molecular association, these results are consistent with Rtn-1C and Spastin sharing a similar subcellular localization, potentially enabling their functional interaction in distal axons” (lines 285-287)

2. We would like to clarify a possible misunderstanding: in our experiments, the increase in microtubule growth rate was observed after axonal Rtn-1 KD. Spastazoline (SPTZ) only prevented the reduction in β 3-tubulin levels induced by Rtn-1 KD, while leaving the KD-driven increase in growth rate and track length unaffected (Figures 5B–E). Thus, our interpretation is that axonal Rtn-1 KD correlates with increased Spastin function. (lines 307-309)

Other comments:

- *Generally, graphs would benefit from individual values plotted as well as the summary. Font sizes and types (but rarely) are sometimes inconsistent. Proteins should be consistently written (capitalised or not).*

R: We agree with the reviewer and thank for taking the time for noticing these inconsistencies as it significantly affects the quality of the work. We have improved several figures and added graphs plotting individual values (Figures: 2 C, 2E; 4 (A-E); 5E; 6D). We have reviewed the Font size and types more carefully and capitalized the proteins accordingly.

- *Table 1 and figure 1 present data collected from a vast amount of resources. It should be highlighted that datasets from which data was obtained includes many different models, different DIVs and neuronal cell types. Figure 1B may benefit from a different colour scheme. "Ex-vivo" should be "Ex vivo". For "ER mRNAs are a relevant category" it is not described what "relevant" would mean in this context. The title might remove this small part or describe it in the text. It should be described how it is decided that mRNAs are "common".*

R: We have now highlighted in the result section the diverse origins of the analyzed samples; We removed the indicated part from the text and explained that common mRNAs were chosen based on the Benjamini–Hochberg (Ben) analysis. (Page 33, lines 1299-1304).

- Figure 2: add description to y-axis to describe what fold change is displayed, applies to multiple figures. Will improve readability of the figures. In 2C, the ROI showing neuronal somata should be increased to show part of the axon and not cut off the soma.

R: We thank the reviewer for taking the time to highlight this. We have included this modification in figure 2 and throughout the article. We have also enlarged the indicated ROIs in figure 2C as requested. (Page 34)

- Figure 3: Three out of four axonal compartments seem to be comprised of dying or damaged axons. Especially the axonal KD scrambled image. It should be ensured that neuronal cultures are healthy.

R: We completely agree with the reviewer that the selected images were not describing the general good health of axons which has been accredited by the lack of fragmentation and functional responsiveness shown in (Figure 4 and 5 B, C, E). Thus, we have now replaced the previous axonal fields by more representative ones (Figure 3). (page 36)

Typo in "intersections". The schematic of 3B is a great addition to explain the graphs above. Perhaps it could be a bit refined as it is currently hard to see whether this is a neuron or a growth cone without context. Maybe show where the axon connects to the depicted growth cones and change the third icon which looks like it was crossed out. Small formatting issues: remove additional space bar before "Figure 3." And add after "Bar"

R: Many thanks for these great suggestions. We have now improved the figures as suggested and changed the indicated formatting issues. (page 36)

- Figure 4: If not misunderstanding what is depicted, in 4A and B, different lookup tables are used to depict the same signal. Only one of each images is necessary. Do the axons have more tiny branches in the Rtn-1 KD condition in 4A? Unclear why Rtn-1 levels are increased in the Rtn-1 KD (4C), please clarify.

R: We thank the reviewer for these observations. The reviewer is correct that different lookup tables were initially applied to the same image. Our intention was to highlight the fine distribution of axonal Rtn-1, but since this aspect is already clearly shown in previous figures, we now retain only a single lookup table. The appearance of tiny branches in the Rtn-1 KD condition represents an isolated observation and does not reflect a consistent or robust phenotype associated with Rtn-1 KD.

As the reviewer points out, the increase of Rtn-1 in the cell bodies of injured neurons following axonal KD was initially surprising to us. However, this was a consistent phenomenon, as shown in the improved Figure 4. Of note, previous studies have reported that total Rtn-1C (but not Rtn-1A) levels increase in response to injury in cortical neurons (Fan et al., 2018). In our case, we interpret this as a compensatory somatic response triggered by the local reduction of Rtn-1 in injured axons. This interpretation is also consistent with the apparent lack of effect of siRNA on distal axonal Rtn-1 levels when applied locally after injury (while somatic application of the same siRNA does reduce axonal Rtn-1). Thus, after 24 hours of KD, the somatic upregulation of Rtn-1 may partially compensate for its expected local synthesis decrease. We have clarified this assumption in the revised text. (lines 247-251)

- Figure 5: It may be easier to understand what "axotomy" samples are if just referred to as "injured" as later in the same figure. The procedure could also very briefly be explained in the results. 5C should depict AUC in μm^2 not μm . 5D Spastin is barely visible, brightness and contrast should be adjusted to enhance visibility.

R: We thank the reviewer for these helpful suggestions and have implemented the requested changes in Figure 5. Specifically:

We now consistently refer to “axotomy” samples as “injured” throughout the figure and article. In addition, a brief explanation of the axotomy procedure has been added before Figure 2 and before figure 5, also the description has been clarified in Materials and methods. (lines 191-192) and (lines 289-290) and (lines 779-787)

To improve the reproducibility of our outgrowth measurements, we revised this analysis approach. Based on previous work from a co-author (McCurdy et al., 2019), instead of reporting the “relative number of intersections,” we now present the total counts obtained from Sholl analysis of binarized axons (see Methods). To this end, we took advantage of the NeuroAnatomy plugin of FIJI, which more precisely tracks axon trajectories and makes the measurement more independent of axon width. Also, this new approach avoids the conflict we had with what we considered the “first line” after the groove ends, which was a bit of arbitrary. Accordingly, the correct term is now “summation of intersections (sum.)” at different distance bins, as reflected in Figure 5D. (page 40)

For the former Figure 5D (now Figure 5B), we have improved the acquisition of representative images and applied a different set of lookup tables to enhance visibility. (page 40)

- Figure 6: It should be made clear why it is necessary to switch to another model system just for 6A, please indicate this in the text. PCR bands seem very pixelated, check the quality. It is unclear why some genes/proteins were only tested with either PCR or WB others with both. Rtn-1C and Rtn1-A should be presented in the same order in the PCR and WB panel. Correct "Rtn1-1A" typo. In 6D, 1.5 dots per soma seems like a low number. When normalized to the area the soma vs the axon occupies, the compartmentalization does not work? Maybe it makes sense to refine analysis or apply puromycin in the somatic compartment and analyze the axonal compartment as comparison?

R: Many thanks for these observations. We have now included the following clarification in the text: “We sought to characterize the isoform expression of Rtn-1 mRNA and protein in both axons and cell bodies. Because microfluidic chambers yield only limited cellular material, we adopted an alternative culture approach using ‘neuronballs.’ This method enables the segregation of an axon-enriched fraction by mechanically separating axons from somato-dendritic structures” (lines 375–376).

The resolution of PCR bands has been improved in the revised figure. Note that because the amount of cellular material is relatively scarce, we did not obtain too strong bands.

The difference in the genes/proteins used for characterizing RNA and protein samples reflects our intention to treat both approaches as complementary. The PCR markers were primarily included to confirm sample purity, which also applies to the WB samples since they derive from the same preparation. In both assays, we used MAP2 as a dendritic marker to demonstrate axonal purity. While we acknowledge that the same genes could have been tested by both methods, we believe the results as presented adequately demonstrate the effective isolation of axons.

We have switched the order of Rtn-1C/1A for consistency across PCR and WB panels and corrected the indicated typo in Figure 6A.

We agree with the reviewer that an average of 1.5 puncta per soma initially appeared low. We have identified at least three reasons for this:

First, the signal derives from only a 15-minute puromycin pulse, which is a very short labeling window. Second, our puro-PLA assay is particularly stringent, as ligation relies directly on puromycin- and Rtn-1C-labeled primary antibodies, without the additional spacing normally introduced by secondary antibodies. In standard PLA, the critical distance for amplification is ~30–40 nm, whereas in our assay this distance is even more restrictive. Third, in our initial analysis we applied an overly cautious threshold to define “true” amplification. We have now refined this threshold using a baseline defined by the absence of puromycin stimulation. With this improved criterion, we now quantify an average of ~5 puncta per soma and ~10 puncta per 1000 μm^2 of axonal area (Figure 6D and Supplementary Figure 3D). Assuming a neuronal soma diameter of 15 μm (area $\approx 176.71 \mu\text{m}^2$), this yields ~0.028 puncta per μm^2 in soma. In comparison, axons display ~0.01 puncta per μm^2 , approximately one-third of the soma value, which is compatible with the idea that cell bodies dominate neuronal protein synthesis.

Following the reviewer’s valuable suggestion, we performed additional quantifications in which puromycin was applied exclusively to the somatic compartment. Under these conditions, we still observed amplification in axons (~5 puncta per 1000 μm^2), although this value was significantly lower than when puromycin was applied directly to axons. This analysis provided a novel appreciation of the puro-PLA technique in neurons: at least half of the signal originates in the axonal compartment, while a portion may reflect proteins synthesized in soma and transported

anterogradely to the axon through yet-unknown mechanisms (potentially involving rapid anterograde transport) (Figure 6D). (page 42)

- *Figure 7: 7A shows two images depicting the same information that may not be needed. Can probably be removed. In 7B there is no negative (or any) correlation between Spastin levels and Tubulin, however later it is mentioned that Rtn-1C transports Spastin thus causing a decrease in Tubulin at certain locations? It is unclear if Spastin levels vary intensely between different samples. Mean intensity of the somatic area may be beneficial to rule this out. 7B Tubulin on the right top panel seems to have a decrease in Tubulin levels which is not visible due to the Y axis of Tubulin being set to a different range than the middle and lower panel. The average of line scans from multiple cells may be helpful to determine whether there is indeed no colocalization between Rtn-1A and Spastin. The provided representative images seem to show similar degrees of colocalization between Spastin and Rtn-1A/C.*

R: We thank the reviewer for these valuable observations and acknowledge that Figure 7 may have caused confusion. We have eliminated the fluorescence line-scan traces, as they can be biased depending on the region of the cell analyzed. Although this may not have been sufficiently emphasized in the text, we had already performed a quantitative colocalization analysis across multiple cells and independent experiments, using Mander's coefficients (Figure 7B). These analyses showed higher colocalization between Rtn-1C and Spastin compared to Rtn-1A. Regarding the concerns about variability in Spastin levels or possible bias from Y-axis scaling, we have eliminated those traces by the risk of bias. Also, we had already quantified the total tubulin fluorescence intensity across all the z-stacks and from multiple cells from independent experiments as shown in Figure 7C. To further rule out artifacts caused by variable transfection efficiency, we quantified total fluorescence intensity in both RFP and GFP channels across conditions. As shown in Supplementary Figure 6, no significant differences were observed, suggesting that the changes in tubulin reflect specific effects of Spastin/Rtn-1C co-expression rather than variability in expression levels.

Results:

- *It would be helpful to reiterate the hypothesis at the start to ease the reading flow.*

R: Many thanks, we have introduced a line reiterating the hypothesis as suggested (lines 117-118)

- *There seems to be minor redundancy in lines 132-138.*

R: Indeed, we have now removed the indicated phrase.

- There are several spellings, proof-reading is recommended. For example, in line 136 should be "promotes". 160 "localla", 192 should be "the actin cytoskeleton", 194 should be "we first examined", 195 should be "Different", 223 "using", 259 "axons".

R: We apologize for the spellings; we have now performed a careful proof-reading and introduced these corrections.

- 154-155: *Unclear, why the lower MW Rtn-1C was seen as more important.*

R: We apologize for not being clear enough. It is not necessarily more important, but we just took the Rtn-1C molecular weight as reference for the analysis considering that this isoform is the predominant in axons. In any case we have found a significant effect for both isoforms at least on siRNA 2 (data not shown), which is now expressed in the text (line 165-169) : “We also examined the 180 kDa band and found that siRNA 1 reduced expression to a mean of 0.41 relative to Scr, showing a strong trend that did not reach statistical significance ($p = 0.05$; $N = 3$; Wilcoxon test compared to 1, data not shown). In contrast, siRNA 2 further reduced expression to a mean of 0.29, which was statistically significant ($p = 0.04$; $N = 3$; Wilcoxon test compared to 1, data not shown).”

- 167 *results of 2E not stated before interpreting them.*

R: We have corrected this mistake.

- 181 *would suggest "outline" instead of "perimeter".*

R: We have considered this suggestion and included “outline”, nevertheless the morphometric parameter is defined as perimeter, so we retained the term, but with the suggested clarification.

- 183-184 *"longest shortest path" is a confusing term.*

R: We agree that it is a confusing term, thus have now introduced multiple clarifications for the term in the legend of figure 3 (page 36), and with more detail in a new section of Materials and methods (lines 697-699).

- figure 4B should be referenced earlier in the sentence.

R: We have corrected the sentence in the text.

- 243-244 *may be correlation. Rtn-1 and Spastin do not necessarily interact so that this result is achieved.*

R: Thanks for the clarification, we are aware that so far in the manuscript the conclusion is not correct, thus now we have stated at the end of the paragraph: “Together, these observations suggest that axonal Rtn-1 KD correlates with higher Spastin microtubule severing” (lines 307-309)

- 246: *In figure 1 the KD seemed to influence both Rtn-1 isoforms, why not here anymore? 259 "axons". 284 "counteract" instead of "suppress"?*

R: We acknowledge the confusion at this point of the article because of measuring a specific isoform. We now indicate that we will focus on Rtn-1C because of previous evidence of the literature pointing to an interaction of Rtn-1C with Spastin (line 264-267). Later we show that Rtn-1C is the predominant isoform in axons (Figure 6). We have corrected all the suggestions in the manuscripts.

- 485: *rephrase as the interaction between Rtn-1C with Spastin has not been shown directly in these experiments.*

R: Many thanks for the relevant clarification. Now, we have corrected:” Here, we have described an emerging mechanism relating Rtn-1C with the activity of Spastin, which is the most frequently mutated isoform in HSP (Hazan et al., 1999; Mannan et al., 2006).” (line 632-634).

Methods: 535 "in PBS". 543 citation error. 689-699 is it necessary to add a gaussian blur?

R: We have corrected the words and removed the wrong reference. Regarding the use of Gaussian blur, it is a very important point. We used this approach because, in our experimental conditions, it was critical to highlight moving particles that otherwise would go unnoticed by the noise. This was particularly manifest for the seemingly more “unorganized” movements of axonal microtubules after injury.

References: Mannan, A U et al. appears twice in the citation list (36 and 44).

R: Many thanks for the observation. Now we have corrected it.

Reviewer #1 (Significance (Required)):

Overall, this manuscript describes novel findings which will be interesting to the neuronal cell biology community and scientists working on the field of neuronal injury and regeneration. It is well structured, and the data are mostly well presented but sometimes conclusions are over-interpreted. However, several points need to be addressed in a more convincing way.

Reviewer #2 (Evidence, reproducibility and clarity (Required)):

Axonal mRNA localization and localized translation support many neuronal functions and is an important determinant of the regenerative potential of axons after injury. How this works mechanistically remains unclear. The authors present a well performed and technically challenging study in which they identify RTN-1 as a regulator of axonal outgrowth after injury. They provide evidence using experiments in microfluidic chambers that RTN1 is locally synthesized in axons. Interestingly, they identify a (local) interplay between RTN1 and Spastin which affects microtubules and thereby regulates the outgrowth of cortical axons after injury. This study provides an interesting new link between a locally synthesized protein (RTN1) and a microtubule-regulating protein Spastin that is changed upon axon injury. This provides an advance in our understanding in axon regeneration after injury and provides the basis for new studies that can further investigate this interplay. Although interesting, I have several concerns that should be clarified and are needed to substantiate the findings and model presented in this study.

Major concerns:

1. *In figure 1, the authors provide an analysis of overlapping axonal mRNAs. There are more axonal transcriptome studies and a recent study by von Kugelgen and Chekulaeva (2020; doi: 10.1002/wrna.1590) already performed such an analysis, which included more studies. It would be good to mention this. It can be perceived that studies were now chosen to get the outcome that Rtn-1 is present in all studies. For example, von Kugelgen finds mRNA coding for RTN3, another ER structural protein, as present in 16 out of 20 studies analyzed. That said, the authors present more reasons to look at Rtn-1, so the selection to continue with this protein remains valid but can be written up differently so not to present it as the 'sole' ER-shaping protein consistently present in axonal transcriptomes.*

R: We appreciate this important observation to enrich the article; we are aware that the transcriptome data can be even further expanded to more recent studies. Thus, we have now included this reference in the main text and highlighted the relevant finding of RTN3. However, Kugelgen and Chekulaeva used data from dendrites/axons (neurites). Thus, we indicate that "...On a similar approach, but combining data from dendrites and axons, it was found that *Reticulon-3* mRNA is present in 16 out of 20 studies, further suggesting a wider presence of other mRNAs coding for ER structural proteins in axons" (line 128-131)

2. *The description of methods is currently insufficient and incomplete and does not allow for reproducibility of this study. For example, different Rtn-1 antibodies seem to be used in this study. Is the same antibody used for staining and WB? There is no listing of any of the antibodies used in the study and which one is used for which technique/experiment. This should be clarified and should be easy to do so in the methods section (antibody name, origin/company, dilution used) to enhance reproducibility of this study. This is not limited to primary antibodies and any information on secondary antibodies, including what was used for STED is completely missing.*

R: Thanks for these critical comments. First, we apologize for the former method version which was mistakenly not as accurate as it should. We have now revisited it and improved several points throughout this section. Regarding the use of primary and secondary antibodies, plasmids, siRNAs, and general reagents, they are all indicated in the Supplementary material, including company and dilution ("Reagent tables").

3. *The timeline of KD experiments in Figures 2 and 3 are unclear. For the Western blot KD is performed at DIV7 and collected 48 hours later. However, this is not specified for the stainings done in Figure 2C-E. Is this also at DIV7 and then for 48 hours? In figure 3 the siRNA is added at DIV8 (together with axotomy) and outgrowth is measured 24 hours later. Is 24 hours sufficient to achieve knockdown? Is this also what was done for stainings? Later on in Figure 5B, 48 hours of KD is again used. It is unclear what the rationale of these differing timepoints is. Why was this chosen? Is the timeline also the reason for the difference in segment lengths chosen? In Figure 3, there is a significant effect on outgrowth in the KD in the 'mid-range' which is not present in Figure 5.*

R: We regret the confusion, now all this information is explicitly clarified in the main text (lines 297-299) and the corresponding figure legends. We have strong reasons to have used these different time points. Figure 2 A-B is aimed at validating the siRNA against Rtn-1 thus we treated 7 DIV cultures for 48 hours to be sure of revealing a global effect by WB. In figure 2 C-D, we used the same 7 DIV cultures, but only for 24 hours. The reason for this is that, once the RNAi was validated, we explored its control on local synthesis in a shorter period based in previous literature supporting that axonal KD for 24 hours is sufficient for regulating axonal transcripts (Batista et al., 2017; Gracias et al., 2014; Lucci et al., 2020). We are also confident of using this time point based in the new supplementary figure 3D that shows a significant decrease on puro-PLA signal (indicative of Rtn-1C synthesis) 24 hours after axonal KD.

In figure 3, we performed axotomy thus we had to wait a longer period for axons to grow (8 DIV) before fully cut them out, in this case we performed axonal KD from 8 to 9 DIVs. This is the same period used for the staining and quantifications shown in figure 4. All this is properly clarified in the main text and figures.

In Figure 5 we performed a more challenging experiment that required to transfect cells with an EB3-GFP plasmid, then perform axotomy along with axonal KD as well as pharmacological treatment selectively in axonal compartment. First, we tried to measure microtubule dynamics under the same temporal frame of figure 3. Nevertheless, expression levels of EB3-GFP were not adequate for axonal measurements by live-cell imaging. Therefore, compared to figure 3, we increased the time frame after axotomy 24 hours (from 9 to 10 DIV) by this technical reason, but also to explore whether the changes on tubulin intensity might be revealed more clearly (which was the case, figure 5B). These considerations are now included in the main text

Regarding the significant effect on outgrowth in the KD in the 'mid-range' which is not present in Figure 5. Given that in figure 5D axons are left growing for two days instead of one, the number of intersections and the differences between conditions is modified compared to figure 3, while retaining the overall trends. Note that to improve the reproducibility of our outgrowth measurements, we revised this analysis approach. Based on previous work of a co-author (McCurdy et al., 2019), instead of reporting the “relative number of intersections,” we now present the total counts obtained from the Sholl analysis of binarized axons (see Materials and methods). To this end, we took advantage of the NeuroAnatomy plugin of FIJI, which precisely tracks axon trajectories and makes the measurements more independent of axon width segmentation. Also, this new approach avoids the conflict we had with what we considered the “first line” after the groove ends, which was a bit of arbitrary. Accordingly, the correct term is now “summation of intersections (sum.)” at different distance bins, as reflected in the new Figure 5D.

4. *Could the authors provide a rescue condition for their siRNA (using a siRNA-resistant construct) to show that their siRNA is specific for RTN1. They nicely show the efficiency of the siRNA but not its specificity. This is crucial because if not specific, this will affect a large part of their study.*

They already have RTN1A and RTN1C constructs available. Such a rescue experiment should ideally also be performed for one or more of their phenotypic experiments, such as the one presented in Figure 3A or 5 to show that the phenotype is really RTN1 dependent. If done by re-expressing either RTN1A or RTN1C, this could provide insightful information on the relevant isoforms.

R: We agree with the reviewer that this is a critical point. A major challenge in demonstrating the functional role of axonally synthesized proteins using a KD approach is that the rescue may also need to occur locally. Since axonal Rtn-1 appears to play a distinct role compared to its somato-dendritic counterpart (Figure 3), a siRNA-resistant construct would ideally require an axon-targeting sequence to restore local synthesis. As this is technically demanding, we have not yet been able to perform such an experiment, but we are actively working on identifying the optimal sequence to direct Rtn-1C to axons. Importantly, studies performing axonal KD typically rely on at least two independent siRNA sequences, thereby minimizing the likelihood that a phenotype arises from off-target effects. Thus, we have now validated a third siRNA (siRNA 3), which selectively downregulates Rtn-1C. Then, following the same experimental frame of figure 3, we performed axonal Rtn-1 KD after injury and observed that siRNA 3 also significantly increases the outgrowth of injured axons (Supplementary figure 2). This suggests that, at least this phenotype, is not product of an off-target effect. Complementarily, pharmacological rescue with the Spastin inhibitor SPTZ mitigated both the reduction in distal axonal β 3-tubulin and the increase on axon outgrowth, supporting that the observed phenotypes are unlikely to arise from off-target effects. If these effects were due to random interference with unrelated mRNA targets, inhibition of an ostensibly independent target such as Spastin would not be expected to yield such a consistent rescue. Accordingly, SPTZ treatment alone did not increase β 3-tubulin, indicating that its action is specifically contingent upon Rtn-1 KD. Taken together, the pharmacological rescue in axons (Figure 5B) and the Rtn-1C/Spastin co-distribution in heterologous cells, which correlates with preserved microtubules (improved Figure 7), provide converging evidence to suggest that Rtn-1C–Spastin interplay may underly the observed phenotypes in axons.

5. I find the data presented in Figure 4A/B confusing. Axonal RTN-1 KD does not reduce axonal RTN1 levels, but somatic KD does. I understand that this implies most protein comes from the soma, and the authors indeed present an explanation that increased somatic RTN1 occurs after axonal KD as a compensation mechanism. However, this can also be interpreted that there is no axonal synthesis of RTN1 after injury and axonal KD has indirect or even aspecific effects. Their model depends on this difference. Their data in Figure 6 could provide supporting evidence if it shows RTN1 puro-PLA after injury. Along these same lines, in Figure 6, they nicely include a compartment control for puro-PLA. It therefore seems doable to include a somatic puromycin control for their axonal puro-PLA, to exclude and diffusion/transport of the newly synthesized peptides. This is especially considering two recent papers reporting on this possible phenomenon, although these studies were not performed in neurons.

R: We consider the possibility that after injury there is no axonal Rtn-1 synthesis as a plausible and relevant appreciation. Unfortunately, we could not perform a puro-PLA experiment after injury,

which would have provided a more definite answer. However, now we are more confident of regulating Rtn-1 synthesis before injury as supported by a new supplementary figure 3D that shows a significant decrease on puro-PLA signal (indicative of Rtn-1C synthesis) 24 hours after axonal KD. Thus, based on the similar phenotypes observed before and after injury, we consider our results are still compatible with Rtn-1 axonal synthesis being downregulated, but not absent after injury. First, axonal Rtn-1 KD decreased β 3-tubulin levels before and after injury according to figure 5B and the improved statistical analysis performed on figure 2E. Similarly, axonal Rtn-1KD significantly increases microtubule growth rate before and after injury according to the current statistical comparisons (Figure 5E). Second, if β 3-tubulin decrease was a merely unspecific siRNA targeting, it is unlikely that SPTZ treatment should increase and restore β 3-tubulin levels only in the context of axonal Rtn-1 KD (Figure 5B). We have now included these considerations in the discussion (lines 537-543). Although on a different track, the mechanistic relationship between Rtn-1C and Spastin suggested in Figure 7 could make more plausible that a similar phenomenon regarding the control of tubulin levels may occur locally in axons.

Following the reviewer's valuable suggestion, we performed additional quantifications in which puromycin was applied exclusively to the somatic compartment. Under these conditions, we still observed amplification in axons (~ 4 puncta per $1000 \mu\text{m}^2$), although this value was significantly lower than when puromycin was applied directly to axons (~ 10 puncta per $1000 \mu\text{m}^2$). This analysis provided a novel appreciation of the puro-PLA technique in neurons: at least half of the signal originates in the axonal compartment, while a portion may reflect proteins synthesized in soma and transported anterogradely to the axon through yet-unknown mechanisms (potentially involving rapid anterograde transport). Note that we revised the criteria for detecting true amplification spots based in staining without puromycin, which increased true amplification numbers. Still, these seemingly low values are compatible with reflecting a limited amount of time (only 15' of puromycin pulse) and the stringent conditions of this experiment in which secondary antibodies were avoided by directly labeling primary ones. This approach makes the classical 30-40nm distance for PLA even narrower, thus reducing signal. In any case, assuming a neuronal soma diameter of $15 \mu\text{m}$ (area $\approx 176.71 \mu\text{m}^2$), this yields ~ 0.028 puncta per μm^2 in somata. In comparison, axons display ~ 0.01 puncta per μm^2 , approximately one-third of the soma value, which makes sense for the expected difference in ribosome density.

6. In Figure 5A the authors find an increased co-localization (RTN1/Spastin) after axotomy. From their images, it seems that the amount of Spastin is hugely increased, which would by default increase the chance of (random) colocalization of RTN1 on Spastin. Could the authors comment on this?

R: Thanks for this relevant and constructive critique. We formerly based our colocalization analysis on deconvolved images. However, after performing several quantifications through different deconvolution parameters, we were not convinced about the robustness of this finding and the performed staining. Thus, we performed a new set of experiments and found that non-deconvolved images from the STED microscope were more informative about the expected tubular morphology of the axonal ER. Thus, we improved figure 5A, and now the main conclusion is just that both proteins are closely distributed in distal axons before and after injury.

7. In figure 5E and 5F, the condition of scr + SPTZ is omitted. What is the reason for this? The explanation of results in these figures is confusing. The authors report a 'clear trend' in increase in comet track length and lifetime upon addition of SPTZ to axonal RTN-1 KD. This is however not significant. The comparisons that are made afterwards are confusing (e.g. increase in comet lifetime of SPTZ in non-injured axons with RTN1 KD compared to Scr+DMSO and KD + DMSO in injured axons). Their conclusion is axonal RTN-1 synthesis in injured axons (see my concern in the points above on this) governs microtubules growth rate beyond Spastin activity yet blocking Spastin activity still completely blocks the effect of KD on outgrowth.

R: We thank this observation and fully agree that the general description provided in figure 5 E wasn't satisfactory. We have re-organized the descriptions of these results and performed more relevant statistical comparisons (lines 338-359). Based on the reviewer observation, we now conclude: "Together, these results suggest that axonal Rtn-1 synthesis controls microtubule dynamics in both non-injured and injured axons, mostly independently of Spastin-mediated microtubule severing." (lines 357-359).

Other/minor concerns:

- *The gene ontology analysis in Figure 1A contains the category 'Endoplasmic reticulum'. In this category are mainly ribosomal proteins. Although in a gene ontology analysis these proteins will be included in this category, it is misleading in this respect since they are just as likely to be coming from cytoplasmic ribosomes. Although it cannot be excluded that these are ER-bound ribosomes, not in the last place because a recent study (Koppers et al., 2024, doi: 10.1016/j.devcel.2024.05.005) found ribosomes attached to the ER in axons, I believe the category should be adapted or at the least clarified in the text.*

R: Many thanks for the suggestion, which is now included in the text. "Note that several of the identified transcripts in the category 'endoplasmic reticulum' code for cytoplasmic ribosomal components, which indeed can be attached to the axonal ER (Koppers et al., 2024) and be locally synthesized in axons (Shigeoka et al., 2019)." (lines 125-128)

- *Is RTN-1C isoform still an ER-shaping protein or rather an ER protein with alternative functions? The final sentence in the abstract makes a statement that a locally synthesized ER-shaping protein lessens microtubule dynamics. Could the authors provide a clearer description and discussion of the evidence in literature for this? RTN1C has been suggested to perform alternative functions in which case the statement that the local synthesis of an ER-shaping protein is important for axonal outgrowth should be adapted.*

R: We agree with the reviewer and are aware that some non-canonical roles of Rtn-1C may partially explain the observed phenotypes. Thus, we have rephrased the last statement of the abstract: "These findings uncover a mechanism by which axonal protein synthesis provides fine control over the

microtubule cytoskeleton in response to injury.”. Also, we have modified the discussion section introducing new references accordingly...” Some studies have pointed to a non-canonical role for Rtn-1C in the nucleus, including DNA binding and histone deacetylase inhibition (Nepravishta et al., 2010, 2012). It is tempting to speculate that these still emerging roles may also contribute to the observed phenotypes. Of note, different axonally synthesized proteins exert transcriptional control in response to injury or local cues (Twiss et al., 2016).” (lines 576-580).

- Is there a difference in RTN1 distribution or levels pre- and post-axotomy?

R: Thanks for the suggestion, with the new analysis we have only found slight reorganization of Rtn-1C and Spastin in distal axons (Figure 5A). We have also included now quantification of their levels and found no significant differences for both proteins (Supplementary figure 4)

- *Line 100/101 states 'the interactome of the axonal ER provides...'. To my knowledge there has been no study looking at the interactome of the axonal ER specifically. Surely axonal ER proteins are known but there is a difference.*

R: We agree with the reviewer that the phrase was misleading, so we rephrased it in the introduction “...Different lines of evidence support that the protein components of the axonal ER may interact with proteins that regulate microtubule dynamics”

- *Typo line 160 'localla'*

R: Thanks for taking the time, we have now corrected it.

- *In Figure S1 B, please add the DIVs to make it clearer what each graph corresponds to. The legend of S1B states different distances from the cell body but the graph shows distances from the tip.*

R: We have now corrected the legend accordingly.

- *Figure 2C, why does $\beta 3$ tubulin decrease in soma, aspecific effect of siRNA?*

R: This was indeed an unexpected finding. However, we do not observe unspecific or global changes in $\beta 3$ -tubulin levels (see Figure 2A and Supplementary Figure 2). Considering our other results linking Rtn-1 to the regulation of the microtubule cytoskeleton, we interpret this decrease as an indirect effect of Rtn-1 depletion rather than an off-target action of the siRNA. Moreover, if the effect were unspecific, both proteins would likely be reduced in the cell body, given that the siRNA was specifically designed to target Rtn-1 as its primary sequence-specific target.

- *What is the rationale on the opposite effect found in outgrowth in Figure 3?*

R: The apparent opposite outcomes observed in Figure 3 — where axonal versus somatic Rtn-1 knockdown leads to divergent effects on axonal outgrowth — can be explained by compartment-

specific environments and isoform distribution. The siRNA targets the conserved RHD region, reducing both Rtn-1A and Rtn-1C. Axons are enriched in Rtn-1C. Thus, axonal KD preferentially reduces Rtn-1C. In contrast, somatic KD reduces both isoforms. Rtn-1A, predominant in cell bodies, may probably engage other signaling pathways (Kaya et al., 2013). Interestingly, it was reported by Nozumi et al. (2009b) that global Rtn-1 depletion reduces axonal outgrowth in developing cortical neurons. This aligns with the notion that somatic KD mimics a more global loss of function, whereas axonal KD reveals a compartmentalized, pro-regenerative effect due to local Rtn-1C regulation. (All the references indicated here are in the main manuscript). These considerations are now included in the discussion (lines 581-593).

- *Missing word 'we' on line 194*

R: We have corrected it.

- *Typo line 629 'witmn h', please proofread the entire manuscript carefully.*

R: We apologize for the spellings, now we have carefully revised the manuscript.

- *Could the authors comment on why, in Figure 7B/C, GFP only is colocalizing with Spastin-RFP? In general, GFP should be diffusive and not display punctate colocalization with Spastin.*

We appreciate the reviewer's comment. Under normal conditions, GFP displays a diffuse cytoplasmic distribution. However, in our experimental setup, we observed punctate GFP signals only in the context of co-expression with Spastin-RFP. This is consistent with prior reports showing that soluble GFP can occasionally be sequestered into late endosomal structures (Sahu et al., 2011), which are also known to harbor the M87 Spastin isoform (Allison et al., 2013; Allison et al., 2019). To rigorously exclude the possibility of unspecific fluorescence crosstalk, we independently acquired each fluorophore channel and confirmed that GFP puncta were genuine and not due to bleed-through (Supplementary Figure 5). Further, cells expressing only GFP or only Spastin-RFP did not show overlapping puncta, and co-expression of GFP with Rtn-1A-RFP did not produce any apparent overlap, indicating that the punctate GFP pattern is specifically associated with Spastin co-expression. Thus, the observed GFP colocalization with Spastin reflects a biological phenomenon potentially linked to the endosomal localization of M87 Spastin, and not an artifact of imaging or fluorophore bleed-through.

Reviewer #2 (Significance (Required)):

Axonal mRNA localization and localized translation support many neuronal functions and is an important determinant of the regenerative potential of axons after injury. How this works mechanistically remains unclear. The authors present a well performed and technically challenging study in which they identify RTN-1 as a regulator of axonal outgrowth after injury. They provide evidence using experiments in microfluidic chambers that RTN1 is locally synthesized in axons. Interestingly, they identify a (local) interplay between RTN1 and Spastin which affects microtubules

and thereby regulates the outgrowth of cortical axons after injury. This study provides an interesting new link between a locally synthesized protein (RTN1) and a microtubule-regulating protein Spastin that is changed upon axon injury. This provides an advance in our understanding in axon regeneration after injury and provides the basis for new studies that can further investigate this interplay. Although interesting, I have several concerns that should be clarified and are needed to substantiate the findings and model presented in this study.

The audience for this study will be mainly basic research in the fields of both axonal protein synthesis and axon regeneration. My expertise is in the field of mRNA localization and local protein synthesis.

Batista, A. F. R., Martínez, J. C., & Hengst, U. (2017). Intra-axonal synthesis of SNAP25 is required for the formation of presynaptic terminals. *Cell Reports*, 20(13), 3085. <https://doi.org/10.1016/J.CELREP.2017.08.097>

Fan, X. xuan, Hao, Y. ying, Guo, S. wen, Zhao, X. ping, Xiang, Y., Feng, F. xue, Liang, G. ting, & Dong, Y. wei. (2018). Knockdown of RTN1-C attenuates traumatic neuronal injury through regulating intracellular Ca²⁺ homeostasis. *Neurochemistry International*, 121, 19–25. <https://doi.org/10.1016/J.NEUINT.2018.10.018>

Gracias, N. G., Shirkey-Son, N. J., & Hengst, U. (2014). Local translation of TC10 is required for membrane expansion during axon outgrowth. *Nature Communications* 2014 5:1, 5(1), 1–13. <https://doi.org/10.1038/ncomms4506>

Lucci, C., Mesquita-Ribeiro, R., Rathbone, A., & Dajas-Bailador, F. (2020). Spatiotemporal regulation of GSK3 β levels by miRNA-26a controls axon development in cortical neurons. *Development (Cambridge)*, 147(3). <https://doi.org/10.1242/DEV.180232>,

Reviewer #3 (Evidence, reproducibility and clarity (Required)):

This manuscript investigates the relationship between the endoplasmic reticulum morphogen reticulon-1 (Rtn-1) and the microtubule severing protein spastin in axons after injury. The main message and conclusion of the paper is that local axonal synthesis of Rtn-1 plays a role in regulating the microtubule severing activity of spastin by interacting with spastin and inhibiting its activity. This mechanism would be important after injury by regulating axonal growth.

The conclusions of the paper are based on the following claims:

1) Rtn-1 is synthesized locally in axons.

2) Specific downregulation in Rtn-1 in axons using microfluidic chambers affects microtubules abundance (measured by beta-3 tubulin) and promotes axon growth after injury.

3) *Inhibition of spastin MT-severing activity with a specific drug rescues the growth effect induced by axonal downregulation of Rtn-1.*

4) *Rtn-1c interacts with spastin-M87 to limit its MT-severing activity in a cellular system upon overexpression.*

Major comments:

1) Evidence that Rtn-1 is synthesized in axons comes from two experiments. Initially, the authors show that Rtn-1 siRNA transfection in the axonal compartment of microfluidic chambers reduces Rtn-1 levels in axons, suggesting that there is some local synthesis. Although this method is very attractive, I am concerned about the statistical analysis. The graphs show bars rather than individual data points from the average of many neurons (about 300). The plots also show the SEM instead of the SD, thus covering all the variability that is inherent in this type of experiment. The statistics are probably not performed on the 3 biological replicates, but consider the individual neurons as N. This is obviously not correct, since neurons in an experiment may all be affected by the same technical problem and are not independent replicates. For this reason, I am a bit skeptical about this quantification. Another problem is that the quantification of the fluorescence intensity of the sample does not take the nuclei into account. Are the nuclei removed for analysis? Are the images single planes? Addressing the quantification issues is crucial also for data in Figure 4, where the authors show a different effect of Rtn-1 axonal KD after injury.

The second experiment is the Puro-PLA in Figure 6D. This experiment shows an average of 1.5 dots of signal per soma, which is a very low level of translation for this compartment where most of the synthesis should be taking place. In the axons, it is not clear how they calculate the axonal area. Again, the number of dots detected is very low and the physiological significance is questionable. A control with a known mRNA translated in axons would be important.

Finally, as an important control, the authors should show the presence of Rtn-1 mRNA by FISH in their experimental system.

R: We appreciate the critical points addressed here as they moved us to improve the quality of the findings. We analyzed cells/axons as statistical units to increase statistical power given the subtle nature of these local changes. We agree with the reviewer that this approach may increase the risk of finding false positives. To address this point, i) we plotted the individual data points and colored them according with the different experimental dates (all the dates showed a similar trend) ii) We indicated SD instead of SEM iii) We analyzed our data using linear mixed-effects models, with experimental date included as a random effect. This approach allows to preserve the granularity and statistical power, while avoiding pseudoreplication. To exclude artifactual changes, we now analyzed the intensity fold change of total fluorescence normalized to Scr. Our former quantifications were based

on the corrected fluorescence intensity used to construct the plot profiles, which could be adding some distortion to the measurements. These changes were applied throughout figures 2 and 4 (pages 34 and 38, respectively). After these new analyses the formerly presented results remain valid.

We thank the reviewer for raising concerns about the quantification of fluorescence intensity in cell bodies. We now specify in Materials and methods that fluorescence intensity analysis of distal axons (always isolated by the microfluidic chambers) and of cell bodies was performed using the wide-field configuration of the microscope. In all the cases, a single (epifluorescent) plane was analyzed to reflect the total fluorescence of a cell or axon. We did not exclude the nuclear region from the quantifications, as this would also remove cytoplasmic signal located above or below the nucleus.

We also understand the concerns about puro-PLA experiments. We agree with the reviewer that an average of 1.5 puncta per soma initially appeared low. We have identified at least three reasons for this. First, the signal derives from only a 15-minute puromycin pulse, which is a short labeling window. Second, our puro-PLA assay is particularly stringent, as ligation relied directly on puromycin- and Rtn-1C-labeled primary antibodies, without the additional spacing normally introduced by secondary antibodies. In standard PLA, the critical distance for amplification is ~30–40 nm, whereas in our assay this distance is even more restrictive. Third, in our initial analysis we applied an overly cautious threshold to define “true” amplification. We have now refined this threshold using a baseline defined by the absence of puromycin stimulation. With this improved criterion, we now quantify an average of ~5 puncta per soma and ~10 puncta per 1000 μm^2 of axonal area (Supplementary Figure 3D). As it is now included in methods, we calculated the axonal area by binarizing β 3-tubulin staining and only counted the true amplification spots inside this region. Assuming a neuronal soma diameter of 15 μm (area $\approx 176.71 \mu\text{m}^2$), this yields ~0.028 puncta per μm^2 in somata. In comparison, axons display ~0.01 puncta per μm^2 , approximately one-third of the soma value which seems more reasonable. This is also compatible with most of Rtn-1C synthesis comes from the cell body.

Unfortunately, we could not be able to perform puro-PLA of other axonally synthesized proteins. Nevertheless, to further validate our puro-PLA signal, we tested the specificity of the Rtn-1C antibody we used for this assay by WB, IF, and Rtn-1 KD (Supplementary figure 3 A-C). In addition, we performed axonal Rtn-1 KD in microfluidic chambers for twenty-four hours, which elicited a significant decrease in puro PLA signal compared to Scr (Supplementary figure 3D). Together, these results strongly indicate that the quantified signal reflects Rtn-1C synthesis. To prove that Rtn-1 mRNA is present in these conditions, we now included a RT-PCR performed on RNA isolated from the somato-dendritic and pure axonal fractions of 8 DIV microfluidic chambers (Supplementary figure 3D). Note that the presence of this mRNA in axons has been supported by several studies, one of them using cortical neurons of similar DIV and cultured in microfluidic chambers (Table I and figure 1).

2) The effects on tubulin following Rtn-1 downregulation in axons is potentially very interesting, but the authors should be careful because it could also mean that the axons are suffering. Can they also stain for other cytoskeletal markers?

R: Regarding this concern, we are aware that in the former Figure 3 we mistakenly selected axonal fields that did not display healthy axons, which was not the dominant trend. This is accredited by the lack of fragmentation and by the functional responsiveness (microtubule dynamics) shown in Figures 4 and 5B, C, E. We have now replaced the previous axonal fields in Figure 3 with more representative axons (healthy), devoid of varicosities and fragmentation (page 37)

3) The results using SPTZ are very interesting and implicate spastin microtubule severing activity in the observed phenotype. In my opinion these experiments however do not prove that "axonal Rtn-1 is indeed promoting the severing of microtubules by spastin", but simply that the blocking spastin activity prevents the appearance of the microtubular phenotype (which appears still with a mysterious mechanism). What happens if they try to stabilize the cytoskeleton by another mean (with taxol for example?). The authors should rephrase this conclusion.

R: We completely agree with the reviewer's appreciation. We now explicitly indicate in the main text that this is (so far in the manuscript) a still correlative phenomenon that suggests an interplay with Spastin activity "...Together, these results suggest that locally synthesized Rtn-1 normally acts to suppress the outgrowth of injured axons, a process that could involve the microtubule-severing activity of Spastin." (lines 321-323). Later in the article, with the improved Figure 7, we further propose that these findings may reflect a causal relationship, although this mechanism has not yet been directly demonstrated in axons.

4) The last experiment (Figure 7) that aims to connect Rtn-1 and spastin function is very artificial, since it is based on overexpression. Why should spastin M87 interact with an ER morphogen? Endogenously it is conceivable that spastin M1 which localizes to the ER would interact with Rtn-1. Moreover, this experiment needs further controls and quantifications. First, it is quite obvious from panel 7C that there is crossover of signal in the two fluorescence channels (see GFP and spastin). Controls need to be shown, where only one of the two fluorescent proteins is expressed, and the specificity of the laser is tested. This experiment is based on only 1 cell shown where co-localisation is detected based on a line that is placed in a specific area of the cell. The effects on the microtubular network needs quantification.

R: We have now improved Figure 7 and added the requested controls to rule out crosstalk as indicated in Supplementary Figure 5 and in the main text. We agree that under normal conditions GFP should display a diffuse cytoplasmic distribution. However, in our experimental setup, we observed punctate GFP signals only in the context of co-expression with Spastin-RFP. This is consistent with prior reports showing that soluble GFP can occasionally be sequestered into late endosomal structures (Sahu et al., 2011), which are also known to harbor the M87 Spastin isoform (Allison et al., 2013; Allison et al., 2019). To exclude the possibility of unspecific fluorescence crosstalk, we independently acquired each fluorophore channel and confirmed that GFP puncta were genuine and not due to bleed-through (Supplementary Figure 5). Further, cells expressing only GFP or only Spastin-RFP did not

show overlapping puncta (arrowheads), and the co-expression of GFP with Rtn-1A-RFP did not produce any apparent overlap, indicating that the punctate pattern of GFP is specifically associated with Spastin co-expression. Thus, we consider that the observed GFP colocalization with Spastin potentially reflects a true phenomenon and not an artifact of imaging or fluorophore bleed-through.

We thank for these observations and apologize for the confusion in the outline of the former figure 7 and the lack of a better description. As the reviewer indicates, one interesting aspect of the M87 isoform is that lacks the ER morphogen domain (so is soluble or cytoplasmic in principle). However, it also harbors endosome and microtubule binding domains which according to previous literature (now included in the main text) may render it a punctate rather than a homogeneous pattern. Also, M87 is the most abundant isoform in the nervous system, particularly at early development. This is the reason why we selected this isoform to test our model. To clarify this point, we based our colocalization analysis in different cells and experimental dates and analyzed all the z-stacks for each cell (see new figure 7B and methods), the intensity plots (now removed) were only for graphical purposes. Similarly, we had already quantified the total tubulin intensity in COS cells based on many cells from different dates and included the sum projections of all the z-stacks from these cells (see new figure 7C). Thus, we removed the intensity profiles as they were clearly misleading (see new figure 7).

We agree that over-expressing constructs may force interactions or co-distribution of proteins. However, in this case, if the observed results were mainly due to over-expression, we should see a similar trend with isoform A as both constructs are under the control of the same strong promoter (CMV) and harbor the same ER morphogen domain (RHD). Nevertheless, the distribution of M87 tightly mirrors Rtn-1C, which is not the case for Rtn-1A. Only as a theoretical prediction, our molecular modeling suggests that Rtn-1C may be associated with Spastin through its microtubule binding domain (Figure 7E). This would suppose that Spastin “decorates” ER-tubules rather than being in the same ER membranous structure. This discrete pattern of Spastin is more coherent with the distribution of both proteins that is now more clearly observed in distal axons by STED super-resolution (new figure 5A). So, despite a bit unexpected, these results suggest a novel interaction mechanism between these two proteins that deserves further validation.

5) What is exactly the model proposed? The title implies that axonal synthesis of Rtn-1 is important during injury, but the data in the paper rather suggest that upon injury the majority of Rtn-1 is not locally synthesized. If the levels of Rtn-1 do not change, why the effect on the microtubules should be specific? Why would a siRNA against Rtn-1 in axons not affect the levels of Rtn-1, but those of tubulin? The authors should be careful, and test other control siRNAs, and Rtn-1 siRNAs, since it is well known even in more simple cellular systems that the toxicity of individual siRNAs can vary greatly.

We consider the possibility that after injury there is no axonal Rtn-1 synthesis as a plausible and relevant appreciation. Unfortunately, we could not perform a puoro-PLA experiment after injury, which would have provided a more definite answer. However, now we are more confident of regulating Rtn-1 synthesis before injury as supported by a Supplementary figure 3D that shows a significant decrease on puoro-PLA signal (indicative of Rtn-1C synthesis) 24 hours after axonal KD. Thus, based on some similar phenotypes before and after injury, we consider our results are still compatible with Rtn-1

axonal synthesis being downregulated, but not fully absent (the mRNA is still detected, as described by Taylor 2009). As such, axonal Rtn-1 KD decreased β 3-tubulin levels before and after injury according to figure 5B and the improved statistical analysis performed on figure 2E. Similarly, axonal Rtn-1KD significantly increases microtubule growth rate before and after injury according to the current statistical comparisons (Figure 5E). In complement, if β 3-tubulin decrease was merely due to unspecific siRNA targeting, it is unlikely that SPTZ treatment should restore β 3-tubulin only in the context of axonal Rtn-1 KD (Figure 5B). Although on a different track, the mechanistic relationship between Rtn-1C and Spastin suggested in Figure 7 could make more plausible that a similar phenomenon regarding the control of tubulin levels could be occurring locally in axons. We have now included these considerations in the discussion (lines 535-543).

To discard off-targets effects, we have now validated a third siRNA sequence (siRNA 3) specifically designed against Rtn-1 and showed that it selectively downregulates Rtn-1C but not β 3-tubulin in cultured cortical neurons. Then, following the same experimental frame of figure 3, we performed axonal Rtn-1 KD after injury and observed that siRNA 3 also significantly increases the outgrowth of injured axons (Supplementary figure 2). This suggests that, at least this phenotype, is not product of an off-target effect. Thus, the pharmacological rescue of β 3-tubulin levels by SPTZ (Figure 5B) and the Rtn-1C/Spastin co-distribution in heterologous cells, which correlates with preserved microtubules (improved Figure 7), provide converging evidence to suggest that Rtn-1C–Spastin interplay may underly the observed phenotypes in axons.

Minor comments:

In Figure 5A, it would be helpful to indicate the border of the axon. The figure is not really convincing.

Following yours and other reviewer comments, we have analyzed a new set of experiments regarding the STED images of non-injured and injured axons. To eliminate the risk of artifactual descriptions, we have avoided deconvolution and worked directly with raw STED images (Figure 5A). Under these conditions, distribution of Spastin and its intensity in distal axons are not modified by injury, nor those of Rtn-1C and Spastin (Supplementary figure 4). Despite these results, data still supports that both proteins are restricted to similar domains subcellular domains before and after injury.

Reviewer #3 (Significance (Required)):

The manuscript uses complex methods to address an interesting cell biological question of relevance to understand axonal growth regulation upon injury. A limitation of the study is the statistical analysis, which triggers some doubts about the reproducibility of the data. Further experiments and the addition of controls would be important to support the claims of the authors.

November 24, 2025

Re: Life Science Alliance manuscript #LSA-2025-03571-T

Dr. Alejandro Luarte
Universidad de los Andes
Medicine
Chile

Dear Dr. Luarte,

Thank you for submitting your previously revised manuscript entitled "Local synthesis of Reticulon-1C lessens the outgrowth of injured axons and Spastin activity" to Life Science Alliance. We have now had a chance to assess the manuscript and the reviews, as well as the prior rebuttal letter.

We agree with the reviewers that this work contains interesting observations that suggest novel regulation of axonal response to injury and will be important for this field. We feel this work should be published in some form, in accordance with our earlier offer of consideration. We are happy to work with you towards this possibility, with the understanding that the manuscript would have to be significantly revised. A revision would not be sent for external review but would be considered at the editorial level with evaluation by an academic editor.

We agree with both reviewers that the work does not offer clear and direct evidence that Rtc-1 is locally synthesized in axons following injury where it then exerts regulatory effects on axon growth. Thus a suitably revised manuscript must reframe these findings. We suggest the results from compartment-specific knockdown and puro-PLA could be framed as observations on Rtc-1 local synthesis without drawing a firm conclusion. The more broad conclusion that its knockdown modulates axon outgrowth after injury (regardless of localization) is already supported as noted by Reviewers.

Another claim that was felt to be insufficiently supported was the direct interaction of Rtc-1 with Spastin. This assertion was also complicated by some observations that were difficult to reconcile (Fig 5C-D).

Overall these claims must be reworked before we can proceed towards publication. While we understand that the text was changed to address reviewer concerns, it should be significantly simplified. Any claims that are asserted but then heavily toned down should be removed. These include those at lines 537-539, 564-573, 602-604, 636-640, and the final two sentences of the abstract, with corresponding changes in the results section accordingly. Overall, the work should present data on local Rtc-1 synthesis as observations without drawing these into a specific working model. We feel the reader can evaluate these data and consider speculation offered in the discussion, recognizing that the full picture is still not clear. LSA gladly publishes observational/descriptive manuscripts, so this reframing does not render this work unsuitable for publication in our view.

We note that some figures have been improved to show individual data points, however this was inconsistently applied across the manuscript. Please revise Figures 3 and 7 to display individual data points for these important observations.

Two smaller issues to note: we agree with Reviewer 1 that a model figure/graphical abstract would be helpful. However we leave this to your discretion. LSA also does not permit claims that refer to data not in the manuscript ("data not shown"). These claims should be removed.

I understand these are major changes that we are requesting. I would be happy to discuss these requests in more detail via email or phone/videoconferencing. Please let me know which option you prefer, if any. See below for how to submit your revised work when ready.

While you are revising your manuscript, please also attend to the below editorial points to help expedite the publication of your manuscript. Please direct any editorial questions to the journal office. When submitting the revision, please include a letter addressing the editorial concerns laid out here.

Thank you for this interesting contribution to Life Science Alliance. We hope that these comments will prove constructive as your work progresses, and we are looking forward to receiving your revised manuscript.

Sincerely,

-- A letter addressing the editorial comments.

B. MANUSCRIPT ORGANIZATION AND FORMATTING:

December 17, 2025

RE: Life Science Alliance Manuscript #LSA-2025-03571-TR

Dr. Alejandro Luarte
Universidad de Los Andes, Chile
Medicine
San Carlos de Apoquindo 2500, Las Condes, Región Metropolitana
Santiago, Metropolitana 7620001
Chile

Dear Dr. Luarte,

Thank you for submitting your revised manuscript entitled "Reticulon-1 synthesis controls outgrowth and microtubule dynamics in injured cortical axons" and for your patience while we evaluated this work. We appreciate your interest in working with the journal towards a path to publication without further reviewer input. We agree that the new title and text accurately reflect the data shown and appropriately position the more speculative claims for which clear mechanistic data is not shown. After considering the new text, we make a few suggestions to further improve clarity for the reader below. We would be happy to publish your paper in Life Science Alliance pending these changes and final revisions necessary to meet our formatting guidelines.

Line 144: Consider framing these observations not as a "model" as this term connotes more direct evidence than is presented here.

Line 253: Please remove the phrase "which suggests a mechanistic relationship between these two proteins".

Line 311: Consider replacing "any" which is not clear in this context, with "none".

Lines 508-509: Because the "shielding effect" is speculative, this sentence would be clearer if edited to: "More specifically, our results are compatible with outgrowth dependent on a potential shielding effect that Rtn-1C may exert over Spastin microtubule severing."

- Please remove the file with supplementary figures, and leave them uploaded separately. Their legends should appear only in the main manuscript file, after the references.
- Please add the X and Bluesky handles of your host institute/organization, as well as your own and/or one of the authors, in our system.
- Please be sure that the authorship listing and order are correct.
- The contributions selected for Andrea Paula Lima do not qualify them for authorship. Please either update the contributions in our system and in the Author Contributions section of the manuscript, or let us know if the author needs to be removed (and added potentially to the acknowledgment section).
- We discourage making data available upon request rather than having it publicly available, but if there is no alternative for privacy or other reasons, please describe what the data are and why they are not public, whom to contact (with a public email address), and conditions for re-use.
- Please use the [10 author names et al.] format in your references (i.e., limit the author names to the first 10).
- Please add your main, supplementary figure, and table legends to the main manuscript text after the references section.
- Please consult our manuscript preparation guidelines <https://www.life-science-alliance.org/manuscript-prep> and make sure your manuscript sections are in the correct order.
- The table should be numbered with Arabic numerals (1, 2, 3, 4); it can be included at the bottom of the main manuscript file or sent as a separate file.
- It is recommended to exclude figures from the manuscript text and upload them separately.
- Please add callouts for Figures 7F and S6 to your main manuscript text.
- Please add the supplementary methods to the methods section of the main manuscript file.

LSA now encourages authors to provide a 30-60 second video where the study is briefly explained. We will use these videos on social media to promote the published paper and the presenting author (for examples, see <https://docs.google.com/document/d/1-UWCfbE4pGcDdcgzcmiuJI2XMBJnxKYeqRvLLrLS08s/edit?usp=sharing>). Corresponding or first-authors are welcome to submit the video. Please submit only one video per manuscript. The video can be emailed to contact@life-science-alliance.org

To upload the final version of your manuscript, please log in to your account: <https://lsa.msubmit.net/cgi-bin/main.plex>

A. FINAL FILES:

B. MANUSCRIPT ORGANIZATION AND FORMATTING:

Thank you for your attention to these final processing requirements. Please revise and format the manuscript and upload materials as soon as you are able.

Sincerely,

Editor response

Dear Editor,

Thank you for your message and for the opportunity to finalize our manuscript for publication in Life Science Alliance. We have implemented all requested text edits (Lines 144, 253, 311, and 508–509) and updated the manuscript and submission files to comply with the journal's formatting requirements. In particular, figures and supplementary figures are uploaded as separate files; all figure and table legends are included in the main manuscript after the references; callouts for Figures 7F and S6 have been added; and supplementary methods have been incorporated into the Methods section. We also updated the references to the first 10 authors et al. format, confirmed the authorship list/order, addressed the authorship/contributions item noted for Andrea Paula Lima, and provided X/Bluesky handles in the system. The Data Availability statement has been revised to describe the datasets, rationale for not fully public deposition, contact information, and conditions for re-use.

Sincerely,
Alejandro Luarte

Point-by-point response

1) Text edits requested: Line 144 (avoid “model” wording)

Change made (revised sentence): “This distribution is compatible with Rtn-1 being present in the dense network of ER tubules characteristically found in growth cones (GCs), an observation well-documented in earlier studies.”

2) Text edits requested: Line 253 (remove mechanistic relationship phrasing)

Change made (revised sentence): “Together, these results indicate that after injury, the levels of Rtn-1 in the somatic and axonal compartments tend to change in a manner that parallels β 3-tubulin.”

3) Text edits requested: Line 311 (“any” → “none”)

Change made (revised sentence): “Local SPTZ application in axons treated with a Scr sequence displayed a trend to increase outgrowth compared to all conditions, particularly in the 1000–1500 μ m segment, but none of these comparisons reached statistical significance.”

4) Text edits requested: Lines 508–509 (make “shielding effect” explicitly speculative)

Change made (revised sentence): “More specifically, our results are compatible with outgrowth dependent on a potential shielding effect that Rtn-1C may exert over Spastin microtubule severing.”

5) Text edits requested: Line 489 (Figure 7F and S6 callout missing)

Callout 1: “This suggests that Rtn-1C can bind the Spastin MTBD (as indicated in Fig. 7F) and may compete with the binding of this domain to microtubules.”

Callout 2: “These effects cannot be attributed to differences in transfection efficiency, as fluorescence intensity of Spastin-RFP and GFP constructs did not significantly differ across conditions (Supplementary Figure 6)”

Formatting + submission checklist

1. We confirm completion of the following items requested in the editorial email:
2. Supplementary figures are uploaded as separate individual files, and the previous combined supplementary-figures file has been removed.
3. Main + supplementary figure legends and table legends are compiled in the main manuscript file after the references (legends only, figures excluded).
4. X (Twitter) and Bluesky handles were added in the system for the institute and authors (see list below).
5. Authorship order/listing was checked and confirmed as correct in the system and manuscript.
6. Andrea Paula Lima authorship/contributions was more accurately specified.
7. Data availability statement updated to explain what is available, why not fully public, contact information, and reuse conditions.
8. References updated to “first 10 authors et al.” format.
9. Manuscript section order checked against LSA guidelines.
10. Tables numbered with Arabic numerals and placed per guidelines (end of main file or separate upload).
11. Figures were excluded from the manuscript text and uploaded separately as recommended.
12. Figure callouts for Fig. 7F and Fig. S6 were added to the main text.
13. Supplementary Methods integrated into the Methods section of the main manuscript file.
14. Listed X handles:

January 5, 2026

RE: Life Science Alliance Manuscript #LSA-2025-03571-TRR

Dr. Alejandro Luarte
Universidad de Los Andes, Chile
Medicine
San Carlos de Apoquindo 2500, Las Condes, Región Metropolitana
Santiago, Metropolitana 7620001
Chile

Dear Dr. Luarte,

Thank you for submitting your Research Article entitled "Reticulon-1 synthesis controls outgrowth and microtubule dynamics in injured cortical axons". We especially appreciate your understanding in executing the text changes that we felt were required to accurately convey the significance of these observations. It is a pleasure to let you know that your manuscript is now accepted for publication in Life Science Alliance. Congratulations on this interesting work which we agree carries important implications for axonal injury and microtubule regulation.

Decision letters, and point-by-point responses will be published alongside the manuscript. If you do want to opt out of having the decision letters and your point-by-point responses displayed, please let us know immediately.

DISTRIBUTION OF MATERIALS:

Again, congratulations on a very nice paper. I hope you found the somewhat unusual editorial process to be constructive and are pleased with how the manuscript was handled. We look forward to future exciting submissions from your lab.

Sincerely,
